# On the Statistical Mechanisms of Distributional Compositional Generalization

Jingwen Fu [1]   Nanning Zheng [1]

## Abstract

Distributional Compositional Generalization (DCG) refers to the ability to tackle tasks from new distributions by leveraging the knowledge of concepts learned from supporting distributions. In this work, we aim to explore the statistical mechanisms of DCG, which have been largely overlooked in previous studies. By statistically formulating the problem, this paper seeks to address two key research questions: 1) Can a method to one DCG problem be applicable to another? 2) What statistical properties can indicate a learning algorithm's capacity for knowledge composition in DCG tasks? **To address the first question**, an invariant measure is proposed to provide a dimension where all different methods converge. This measure underscores the critical role of data in enabling improvements without trade-offs. **As for the second question**, we reveal that by decoupling the impacts of insufficient data and knowledge composition, the ability of the learning algorithm to compose knowledge relies on the compatibility and sensitivity between the learning algorithm and the composition rule. In summary, the statistical analysis of the generalization mechanisms provided in this paper deepens our understanding of compositional generalization, offering a complementary evidence on the importance of data in DCG task.

## 1. Introduction

Compositional Generalization (CG) represents the capacity to comprehend novel combinations of familiar concepts, an intellectual feat widely regarded as a pivotal milestone in human cognitive evolution (Pearl & Mackenzie, 2018; Harari, 2014). This remarkable ability empowers humans to generate an infinite array of ideas and constructs from finite building blocks of knowledge. For example, humans can understand the concept of the "red triangle" after grasping the "red rectangle" and "blue triangle". To mimic human abilities, this paper explores the question whether machines can generalize to new data distributions that require recombining knowledge from previously learned distributions. For example, this involves generalizing to the distribution of a "red triangle" after learning about "red rectangles" and "blue triangles." We refer to this type of generalization as Distributional Compositional Generalization (DCG).

Nevertheless, machines have consistently struggled to emulate this level of compositional generalization, as it fundamentally challenges the prevalent assumption of independent and identically distributed (IID) between training and test data, a cornerstone principle in the machine learning literature (Kawaguchi et al., 2017; Bartlett & Mendelson, 2002; Bousquet & Elisseeff, 2002; Mohri et al., 2018; McAllester, 1998; Fu & Zheng, 2023; Fu et al., 2023). When faced with the data significantly divergent from the training (support) distribution, achieving meaningful generalization becomes virtually insurmountable (Koh et al., 2021; Sagawa et al., 2021; Dong & Ma, 2022). This stark reality underscores the critical need for a rigorous theoretical examination of DCG, as it holds the key to bridging the gap between human-like adaptability and the limitations of current machine learning models in handling unforeseen, complex combinations of concepts.

A dominant theoretical approach to understanding the generalization properties of learning systems is the statistical method. Over the decades, numerous statistically-based methods have been developed to enhance our understanding of generalization behavior under the IID assumption within the PAC learning framework (Vapnik et al., 1998; Vapnik, 1999). A key aspect of these approaches is the use of statistical methods to formulate tasks and generalization mechanisms. This shift allows for a focus on the statistical properties common across various problems, rather than on specific problems themselves. However, similar results have not been achieved in the area of Distributional Compositional Generalization (DCG). Although various explorations of DCG have been conducted, these methods often take different perspectives, such as identification (Wiedemer et al.,

[1]National Key Laboratory of Human-Machine Hybrid Augmented Intelligence, National Engineering Research Center for Visual Information and Applications, and Institute of Artificial Intelligence and Robotics, Xi'an Jiaotong University. Correspondence to: Nanning Zheng <nnzheng@mail.xjtu.edu.cn>.

*Proceedings of the 42$^{nd}$ International Conference on Machine Learning*, Vancouver, Canada. PMLR 267, 2025. Copyright 2025 by the author(s).

2024; 2023) and group invariance (Ito et al., 2022; Lee et al., 2024). The statistical properties and constraints of DCG remain unclear. In this work, we aim to explore the statistical properties of DCG. Our research focuses on two main questions:

Q1: *Can a method for one DCG problem be useful for another DCG problem, and under what circumstances is a general method applicable to all DCG problems?*

Q2: *What statistical properties can indicate a learning algorithm's capacity for knowledge composition in DCG tasks?*

**For Q1**, we propose an invariant measure that indicates the existence of a dimension where all different methods for DCG are equivalent. This measure helps us understand the mechanisms behind both trade-off and non-trade-off improvements. We demonstrate the critical role of method adaptivity to data in achieving non-trade-off improvements, which supports the importance of a data-centric approach (Zha et al., 2023) in the DCG problem. **Regarding Q2**, we present a new generalization bound for the generalization error in DCG tasks. By decoupling the effects of insufficient data and knowledge composition, our bound demonstrates that the ability of the learning algorithm to compose knowledge depends on two key factors: 1) the compatibility between the learning algorithm and the composition rule (Definition 5.1), and 2) the influence of the composition rule on the algorithm's output, as measured by mutual information.

In summary, the key contribution of this paper is providing a statistical analysis of the generalization mechanisms in the DCG problem, offering a complementary perspective to prior research. Specifically, it provides an invariant measure and explores the relationship between the learning algorithm and the composition rule.

## 2. Related Works

**Statistic Generalization Theory**  Statistical generalization theory (Vapnik et al., 1998; Vapnik, 1999) is a subfield of statistical learning theory that seeks to understand the mechanisms of generalization from a statistical perspective. In this context, data is typically modeled under the IID assumption, meaning that both the training and test data are assumed to come from the same independent and identically distribution. The theory explores generalization mechanisms through the central limit theorem, often within the PAC-learning framework. In this framework, the learning process involves finding a function from a function space that fits the training data. Due to the central limit theorem (Rouaud, 2013), a simpler function space tends to result in a smaller gap between the training error and the test error, known as the generalization error. Therefore, a key

challenge in this theory is to determine an effective measure of the complexity of the function space. Several methods have been proposed for this purpose, such as VC dimension (Vapnik & Chervonenkis, 2015), Rademacher complexity (Bartlett & Mendelson, 2002), and covering number (Shalev-Shwartz & Ben-David, 2014). Besides the complexity of function space, researchers also explore algorithm-based methods, such as algorithm stability (Bousquet & Elisseeff, 2002; Hardt et al., 2016) and information-theoretic analysis (Xu & Raginsky, 2017; Russo & Zou, 2016), to understand generalization. These approaches, like the previous ones, rely on the IID assumption to model data and use statistical laws to understand the generalization mechanism. However, these theories don't fully address the DCG problem, which violates the IID assumption. In this paper, we aim to fill this research gap by proposing a theory applied to DCG, using statistical methods to model data and generalization mechanisms.

**Distributional Compositional Generalization (DCG)** DCG is a subfield within compositional generalization that has garnered significant attention in recent years. Here, we provide a brief overview of both the applications and theoretical research related to DCG. **From an application perspective**, DCG is crucial for addressing unseen scenarios and mitigating the issue of data scarcity. For instance, in text-to-image generation (Liu et al., 2022; Okawa et al., 2024; Li et al., 2024; Du et al., 2023), solving the DCG problem enables the creation of entirely novel images, such as generating an image of a red panda. Even though red pandas don't exist, we can infer this image by combining the distributions of different pandas and animals with red coloration. Similar applications include content and style generalization (Jing et al., 2019; Jin et al., 2022), among others. In reinforcement learning (Silver & Ciosek, 2012; Li et al., 2021; Sutton et al., 1999; Tasse et al., 2022; Bacon et al., 2017), we can only collect the data of the preliminary task and model can solve more complex task by composite. However, without the DCG ability, we have to collect all possible combination of the tasks. **From a theoretical perspective**, researchers focus on understanding the mechanisms for solving DCG problems. Various mechanisms have been analyzed, including disentanglement (Lippl & Stachenfeld, 2024; Wang et al., 2022; Bengio et al., 2013), identifiability (Wiedemer et al., 2024; 2023), and others (Ito et al., 2022; Lee et al., 2024). However, the statistical properties of DCG and its constraints remain largely unexplored. This paper seeks to address this gap by analyzing the statistical properties of DCG, an area that has been mostly overlooked in previous research.

# 3. Problem Definition

## 3.1. Preliminary

**Notations** In this paper, we employ $P$ to signify the distribution and $P(\cdot)$ to denote its corresponding density. Bold symbols represent random variables, while unbold symbols represent their corresponding values. For a random variable $\boldsymbol{x}$, $P_{\boldsymbol{x}}$ represents its distribution. The calligraphic font is used to denote the space or learning algorithm. $I(\cdot;\cdot)$ denotes the mutual information. And $\mathbb{E}$ denotes the expectation.

**Sample Space and Distribution** In this analysis, we consider a data space $\mathcal{Z}$, which can be decomposed into two parts in some case, i.e. $\mathcal{Z} = \mathcal{X} \times \mathcal{Y}$, where $\mathcal{X}$ and $\mathcal{Y}$ are two spaces. We use the notation $\boldsymbol{z}$ to denote a random variable that takes value in the space $\mathcal{Z}$. The distribution of this random variable is represented by the notation $P_{\boldsymbol{z}}$.

**Function space** The function space is denoted by the symbol $\mathcal{F}$, where $f : \mathcal{Z} \to \mathbb{R}_+ \in \mathcal{F}$. The function $f$ assigns loss to the corresponding data point. Given the data distribution $P_{\boldsymbol{z}}$, the error is denoted as $err(P_{\boldsymbol{z}}, f) = \mathbb{E}_{\boldsymbol{z} \sim P_{\boldsymbol{z}}} f(\boldsymbol{z})$. Similarly, the corresponding error can be expressed as $err(D_n, f) = \frac{1}{n} \sum_{z \in D_n} f(z)$ for the finite samples $D_n \sim P_{\boldsymbol{z}}^{\otimes n}$. For supervised learning, where $z = (x, y)$, we can decompose the function $f$ as $f = l(h(x), z)$, where $l$ is the loss function and $h$ is the function that maps the input to its corresponding prediction.

**Learning algorithm** Given the function space, the role of the learning algorithm is to find suitable functions for the given problem. Here, we denote the learning algorithm as $\mathcal{A} : \mathcal{D} \to \mathcal{PF}$, where $\mathcal{D}$ denotes the space of all training data and $\mathcal{PF}$ denotes the space of all distribution on the function space. The output of the learning algorithm is regarded as a distribution over the function space, rather than a single function because the learning algorithms typically encompass a degree of uncertainty, for instance, stochastic noise in optimization. We denote the operation on dataset $D_n$ as $\mathcal{A}(D_n)$, and similarly, we denote the operation on the infinite data sampled from $P_{\boldsymbol{z}}$ as $\mathcal{A}(P_{\boldsymbol{z}})$. In this paper, the learning algorithm includes not only the optimizer (e.g. SGD, Adam), but also any constraints or techniques that influence the selection of functions from the function space.

**Induced Distribution** Given the learning algorithm $\mathcal{A}$, we use $Q^{(\mathcal{A})}$ to denotes the distribution induced by the learning algorithm $\mathcal{A}$. Given a data distribution $P_{\boldsymbol{z}}$, we denote $Q_{\boldsymbol{f}|P_{\boldsymbol{z}}}^{(\mathcal{A})} \triangleq \mathcal{A}(P_{\boldsymbol{z}})$ and the corresponding density is denoted as $Q_{\boldsymbol{f}|P_{\boldsymbol{z}}}^{(\mathcal{A})}(f|P_{\boldsymbol{z}})$ and the random variable as $\boldsymbol{f}$. We will drop the subscript $\boldsymbol{f}|P_{\boldsymbol{z}}$ if no ambiguity caused.

## 3.2. Compositional Distributions

**Subdistribution** We denote the two compositional components as $a \in A$ and $b \in B$. The other components, including randomness, are denoted as $\zeta$. The overall data distribution $P_{\boldsymbol{z}}$ can be divided into several subdistributions based on the different values of the compositional components. These distributions are $\{P_{a,b}\}_{a \in A, b \in B}$. The $P_{a,b}(z)$ satisfies that $P_{a,b}(z) = \frac{P_{\boldsymbol{z}}(z)}{|A| \times |B|} \mathbf{1}_{(a_z = a) \wedge (b_z = b)}$, where $a_z$, $b_z$ are the corresponding fact values of the sample $z$. To ensure that each sample has only one determined factor value for each factor, the distribution should satisfy that for any $a_1, b_1, a_2, b_2$, where $a_1 \neq a_2$ or $b_1 \neq b_2$, we have supp $P_{a_1, b_1} \cap$ supp $P_{a_2, b_2} = \emptyset$.

**Distribution split** We denote $E = A \times B$ as the all possible combinations of the component $a, b$. We define $S, U$ as a partition of the set $E$, i.e., $U \cap S = \emptyset$ and $U \cup S = E$. Based on the partition, the distribution can be divided into the support distribution $P_S = \{P_{a,b}\}_{(a,b) \in S}$ and target distribution $P_U = \{P_{a,b}\}_{(a,b) \in U}$. We denote $err(P_S, f) = \mathbb{E}_{(a,b) \in S}[err(P_{a,b}, f)]$ and similarly for $err(P_U, f)$. We denote $\boldsymbol{f}_S$ as the random variable sampled from $\mathcal{A}(P_S)$ and the same for $\boldsymbol{f}_U$ and $\boldsymbol{f}_E$.

**Remark 3.1.** In this paper, we primarily focus on DCG involving two components. The analysis of two-component DCG serves as a fundamental basis for addressing more complex DCG issues. (Dong & Ma, 2022; Wiedemer et al., 2024; Ren et al., 2024; Petrache & Trivedi, 2024; Chomsky, 2002; Partee et al., 1995; Gordon et al., 2019; Silver & Ciosek, 2012; Li et al., 2021; Sutton et al., 1999; Tasse et al., 2022; Bacon et al., 2017; Wiedemer et al., 2023; Brady et al., 2023).

In the following, we give several examples:

**Example 3.2.** *(Image) In the context of single object images, let $A$ denote the shape of the object and $B$ its size. We define $P_{a,b}$ as the distribution of images of shape $a$ and size $b$.*

**Example 3.3.** *(Robot) We consider a task distribution for robots consisting of a walking task and an operational task. Let set $A$ denote a sequence of walking subtasks, while set $B$ denotes a collection of operational subtasks. We define distributions such as $P_{a_1, b_1}$ for slow walking and object picking tasks, $P_{a_2, b_1}$ for regular walking and object picking tasks, and $P_{a_1, b_2}$ for slow walking and object stacking tasks. The target distribution, labelled $P_{a_2, b_2}$, is specifically tailored for tasks involving slow walking and object stacking.*

**Remark 3.4.** Certain DCG tasks require models to master basic components before progressing to more complex challenges. Take robot learning as an example: initial tasks may focus only on activities such as walking and retrieving objects independently. Subsequently, the network needs to extend its understanding to situations where the robot

performs both activities simultaneously. In such cases we introduce the null component $\emptyset$. The distribution related to a single component can be represented as $\mathbb{P}_{a,\emptyset}$ or $\mathbb{P}_{\emptyset,b}$. We can set $A' = A \cup \emptyset$ and $B' = B \cup \emptyset$.

### 3.3. Distributional Compositional Generalization

In this section, we set out to formulate the DCG. The relationship of the DCG problem can be summarised in the diagram:

$$
\begin{array}{ccccccc}
T & \longrightarrow & P_S^{(T)} & \xrightarrow{\mathcal{A}} & \boldsymbol{f}_S & \xrightarrow{\beta} & \tilde{T} \\
& \searrow & & & \downarrow{\scriptstyle err} & & \\
& & P_U^{(T)} & \xrightarrow{err} & err(P_U^{(T)}, \boldsymbol{f}_S) & &
\end{array}
\tag{1}
$$

**Composition Rules and Data Generation**    For any two different distributions, $P_{e_1}, P_{e_2} \in P_E$, there exists a composition rule $T$ that connects them. This composition rule acts as a bridge for these different distributions to become part of a problem. We define a generation function $g(\cdot)$ such that $P_E^{(T)} = g(T, \xi)$, where $(T)$ is used to emphasise that the distribution is generated by the composition rule $T$, and $\xi$ refers to all the other information needed to generate the distribution. For example, if we consider the shape and color composition of an object, then $T$ contains the shape and color components and their composition method. The $\xi$ represents the information other than shape and color, such as position. Usually the $\xi$ follows a certain distribution $P_{\boldsymbol{\xi}}$. We denote this as $\boldsymbol{P}_E^{(T)} = g(T)$, where the $\xi$ is omitted to indicate that it is randomly sampled from the distribution $P_{\boldsymbol{\xi}}$.

**Example 3.5.** *In image creation, the composition rule can be represented as (set of shapes, set of colors, "Draw the contour of a <shape> and fill it with <color>."). Similarly, for a robotic task, the composition rule can be expressed as (set of subtask1, set of subtask2, "First, complete <subtask1>, then <subtask2>."). Keep in mind that we use text to describe the compositional rule here, but it can take any form as long as it defines how two components are combined.*

**Function Space and Learning Algorithm**    Typically, the method for solving a machine learning problem involves two options: 1) design a function space that is more suitable for the given problem, and 2) design a better learning algorithm. In this paper we assume that we are given a large and fixed function space that contains almost all possible functions. We can consider the design of a suitable function space as a hard constraint on the learning algorithm. More specifically, this hard constraint means that only part of the function space can be a legal output of the learning algorithm, although the learning algorithm operates on the rather large function space.

## 4. Invariant Measure

When tackling a problem, it's common to wonder what the approach will entail. The central question is whether a universal method can be applied to a range of tasks or if specialized methods are required for each. To determine this, we must explore how methods for different tasks relate to one another. If a method that works well for one task also proves effective for others, a general approach might be feasible. However, if a method succeeds in one task but fails in others, it becomes essential to develop task-specific strategies rather than relying on a single general method. This section will propose an invariant measure to answer this question.

### 4.1. Analysis

In this section, we aim to analyze the learning algorithm's ability to predict the correct composition rule. However, a gap exists because the outputs of the learning algorithm lie in the function space, while the composition rules reside in a different space. To bridge this gap, we introduce the function $\beta(\cdot)$, which connects these two spaces and allows us to make this comparison.

**Definition 4.1.** We consider a rule prediction function $\beta : \mathcal{F} \to \mathcal{T}$, such that for any function $f \in \mathcal{F}$, we have $\tilde{T}_f = \beta(f) \in \mathcal{T}$.

**Remark 4.2.** Normally we expect that $\beta(\cdot)$ can satisfy the condition that for all $f(\cdot)$, $err(f, P_E^{(\tilde{T}_f)})$ is a small value. However, the choice of $\beta(\cdot)$ is not important in this paper, since the following theorem holds for all $\beta(\cdot)$, as long as their output is a valid composition rule in space $\mathcal{T}$. When $\beta(\cdot)$ operate on the random variable $\boldsymbol{f}$, we can obtain another variable $\tilde{\boldsymbol{T}} = \beta(\boldsymbol{f})$. Its correspoding distribution is denoted as $Q_{\tilde{\boldsymbol{T}}}^{(\mathcal{A},\beta)}$. The subscript is omitted if no ambiguity caused.

The learning algorithm identifies the composition rule based on two mechanisms: the inherent bias of the learning algorithm and the adaptivity of the learning algorithm. The first refers to the learning algorithm's preference for one composition rule over another, and the second refers to the learning algorithm's ability to adjust its predictions in response to the data provided. This leads us to the following research question:

*How can these two mechanisms be modelled and combined in the statistical framework?*

To give an analysis, we first need a mathematical modelling of these two mechanisms:

1) **Inherent Bias**    First, we represent the bias of the learning algorithm as $Q^{(\mathcal{A},\beta)}(\tilde{T})$, which is the marginal distribution over all possible training data. The $\tilde{T}$ indicates the

prediction of the composition rule using $\beta(\cdot)$. This formulation is appropriate because the prediction is not conditioned on any specific data. For two composition rules $T_1$ and $T_2$, if $Q^{(\mathcal{A},\beta)}(\tilde{T} = T_1) > Q^{(\mathcal{A},\beta)}(\tilde{T} = T_2)$, we say that $\mathcal{A}$ is biased towards $T_1$ over $T_2$.

**Remark 4.3.** Inductive bias refers to a model's inherent preference for certain compositional rules before it is exposed to any training data for a given task. This bias can be introduced in two primary ways: 1) Model Architecture Design: By carefully structuring the model, we can constrain its outputs to adhere to specific compositional rules. 2) Pretraining and Objective Function: The inductive bias can also be shaped through pretraining strategies or the choice of objective function, either suppressing or reinforcing the model's tendency toward certain compositional behaviors.

2) **Adaptivity**    The second is the adaptivity of the learning algorithm, which refers to its ability to adjust its predictions in response to the data provided. Based on the definition, an intuitive method is to represent it as $I_{\mathcal{A},\beta}(\tilde{\boldsymbol{T}} = \boldsymbol{T}; \boldsymbol{P}_S^{(T)})$. We denote this as $I_{\mathcal{A},\beta}(\cdot; \cdot)$ because the calculation of mutual information relies on $Q^{(\mathcal{A},\beta)}$, which is influenced by the learning algorithm $\mathcal{A}$ and the rule prediction function $\beta$.

### 4.2. Theorem

Based on the previous statistical formulation, we go into the definition of the invariant measure. Invariance means that this measure is the same for different learning algorithms. As a result, it can serve as a tool for analyzing both the trade-offs and non-trade-off improvements between different tasks.

**Definition 4.4.** Given the composition rule $T$, the distribution $P_S$, the learning algorithm $\mathcal{A}$ and the rule prediction function $\beta$, and a function $\alpha_{\mathcal{A},\beta} : \mathcal{T} \times P \to \mathbb{R}^+$, we define the $\mu$ measure as

$$\mu_\beta(T, P_S^{(T)}, \mathcal{A}) = \frac{Q^{(\mathcal{A},\beta)}(\tilde{T} = T | P_S^{(T)})}{\alpha_{\mathcal{A},\beta}(T, P_S^{(T)})}, \qquad (2)$$

where $\tilde{T} = \beta(\tilde{f}_S)$ ($\tilde{f}_S$ is the prediction made by the Leanring algorithm) and $P_S^{(T)}$ is the support distribution generated by the composition rule $T$.

With the $\mu$ measure defined above, we provide the invariant property of this measure with respect to different methods:

**Theorem 4.5.** *(Invariant Property) There exists at least one function $\alpha$ such that for any $\beta$, the $\mu$-measure satisfies the following conditions:*

- *(1) For any $\boldsymbol{T}$, $\boldsymbol{P}_S^{(T)}$ and $\mathcal{A}$, we have $\mathbb{E}_{\boldsymbol{T}, \boldsymbol{P}_S^{(T)}} \log \alpha_{\mathcal{A},\beta}(\boldsymbol{T}, \boldsymbol{P}_S^{(T)}) = I_{\mathcal{A},\beta}(\tilde{\boldsymbol{T}} = \boldsymbol{T}; \boldsymbol{P}_S^{(T)});$*

- *(2) For any $\mathcal{A}_1, \mathcal{A}_2$, the following equation holds*

$$\mu_\beta(\mathcal{A}_1) = \mu_\beta(\mathcal{A}_2), \qquad (3)$$

*where $\mu(\mathcal{A}) = \sum_T \mathbb{E}_{P_S^{(T)} \sim g(T)} \mu(T, P_S^{(T)}, \mathcal{A})$.*

### 4.3. Discussion

In the following, *we refer the $\alpha$ as the one that satisfies the invariant property listed in Theorem 4.5.* We rewrite the definition of the $\mu$-measure as follows,

$$\begin{aligned} \mu_\beta(\mathcal{A}) &= \sum_T \mathbb{E}_{P_S^{(T)} \sim g(T)} \mu(T, P_S^{(T)}, \mathcal{A}_1) \\ &= \sum_T \mathbb{E}_{P_S^{(T)} \sim g(T)} \frac{Q^{(\mathcal{A},\beta)}(\tilde{T} = T | P_S^{(T)})}{\alpha_{\mathcal{A},\beta}(T, P_S^{(T)})} \qquad (4) \\ &= constant. \end{aligned}$$

Recall that if we obtain the composition rule, then we can reconstruct the ground truth distribution $P_E^{(T)}$. Therefore, the composition rule is the core of our concern in the DCG problem. $Q^{(\mathcal{A},\beta)}(\tilde{T} = T | P_S^{(T)})$ is the probability that the learning algorithm will predict the correct composition rule. Based on this, we use the probability $Q^{(\mathcal{A},\beta)}(\tilde{T} = T | P_S^{(T)})$ as a measure of performance.

**Trade-off**    Recalling the definition of the DCG task in section 3.3, we can specify a DCG task with the $T, P_S^{(T)}$. In this sense, the calculation of $\mu_\beta(\mathcal{A})$ can be seen as an aggregation of the value of the $\mu$ measure across different tasks. If $\alpha_{\mathcal{A},\beta}(T, P_S^{(T)})$ is fixed, we can get a clear trade-off between performance on different tasks. This can be achieved by choosing a different learning algorithm but with the same value of $\alpha_{\mathcal{A},\beta}(T, P_S^{(T)})$ for different tasks. In this situation, increasing the performance of one task with non-zero $\alpha_{\mathcal{A},\beta}(T, P_S^{(T)})$ will result in decreasing the performance of other tasks.

**Beyond Trade-off**    Then we come to the other problem, which is how to improve performance on one task without sacrificing performance on other tasks. The iniuition is that if we can improve the performance by fixing its corresponding $\mu$-measure fixed. Recall the definition of the $\mu$ measure that $\mu_\beta(T, P_S^{(T)}, \mathcal{A}) = \frac{Q^{(\mathcal{A},\beta)}(\tilde{T}=T|P_S^{(T)})}{\alpha_{\mathcal{A},\beta}(T, P_S^{(T)})}$. So we need to increase $\alpha_{\mathcal{A},\beta}(T, P_S^{(T)})$ and $Q^{(\mathcal{A},\beta)}(\tilde{T} = T | P_S^{(T)})$ at the same rate. In this way, we improve performance without altering the $\mu$ measure, making it a non-trade-off improvement. Moreover, Based on the equation that $\mathbb{E}_{\boldsymbol{T}, \boldsymbol{P}_S^{(T)}} \log \alpha_{\boldsymbol{T}}(\mathcal{A}, \boldsymbol{P}_S^{(T)}) = I_{\mathcal{A},\beta}(\tilde{\boldsymbol{T}} = \boldsymbol{T}; \boldsymbol{P}_S^{(T)})$, we can conclude that the statistical dependence $I_{\mathcal{A},\beta}(\tilde{\boldsymbol{T}} = \boldsymbol{T}; \boldsymbol{P}_S^{(T)})$ plays an important rule in non-trade-off improvement.

**Implication for practice** The analysis suggests that developing methods adaptable to data, with careful data engineering, is a promising approach for effectively solving DCG. The relationship between non-trade-off improvements and $I_{\mathcal{A},\beta}(\tilde{\boldsymbol{T}} = \boldsymbol{T}; \boldsymbol{P}_S^{(T)})$ highlights the inevitable sensitivity of data to methods applicable across various tasks. As a result, careful data engineering is essential, supporting the data-centric AI approach (Zha et al., 2023).

**Compared with previous studies.** 1) **Research in DCG.** Previous work (Dong & Ma, 2022; Dziri et al., 2024) has discussed how a method that is effective for one task may struggle to solve another. While these studies are similar to ours in exploring the relationship between a method's performance on different tasks, they primarily focus on DCG problems with specific composition rules. In contrast, our work proposes an invariant measure that reveals the underlying mechanism and is applicable to a wider range of situations, as it considers the relationships between all different composition rules. 2) **Compare with No free lunch theorems.** Another well-known theorem addressing the trade-off between a method's performance across different tasks is the "No Free Lunch" (NFL) theorem, which primarily focuses on problems in optimization (Wolpert & Macready, 1997), search (Wolpert et al., 1995), and supervised learning (Sterkenburg & Grünwald, 2021; Wolpert, 2021; 2002). Further details on the NFL theorem can be found in references (Adam et al., 2019; Joyce & Herrmann, 2018; Ho & Pepyne, 2001; Ho et al., 2003; Yang, 2012; Rowe et al., 2009). The NFL theorem states that, when averaged across all possible problems, any two methods are essentially equivalent. The **main difference** between our study and the NFL framework lies in two key areas: 1) Our work focuses on the composition rule $T$, and our theorem is not limited to any specific learning problem, such as supervised learning (Wolpert & Macready, 1997; Wolpert et al., 1995) or unsupervised learning (Sterkenburg & Grünwald, 2021; Wolpert, 2021; 2002). As long as the problem falls within the DCG category (defined in Section 3.3), our theory applies. 2) While the NFL theorem discusses trade-offs between tasks, our theory highlights non-trade-off improvements specifically within the DCG problems.

## 5. Statistic Mechanism of Knowledge Composition

In the previous section, we are concerned with the situation over all possible composition rules. However, composition rules have different probability of occurring in certain scenarios. Therefore, in this section we consider the problem that the composition rule follows a certain distribution $\boldsymbol{T} \sim P_{\boldsymbol{T}}$ and $\boldsymbol{P}_S^{(T)} \sim g(\boldsymbol{T})$. $D_n$ is the data set sampled from the distribution $\boldsymbol{P}_S^{(T)}$. By modifying the distribution $\boldsymbol{P}_{\boldsymbol{T}}$, we can emphasize the composition rules that are most

likely to occur in practice.

### 5.1. Decouping the influence of insufficient data

Unlike the generalization analysis in the IID assumption, the generalization error of the DCG comes from two sources: The first one is due to the insufficient data and the second one is due to the fairness of combining the prior knowledge to handle the new situation. We refer to the first as the IID error and the second as the CG error. This leads us to the research question:

*How to seperate the influence of insufficient data and lack of knowledge composition on generalization error?*

To make such an analysis possible, we first need to identify the influence of the insufficient data, and then we need to model this influence in a mathematical way.

**1) Evaluation.** Given a function $f$, the generalization error refers to the gap between the error on the target distribution, $err(P_U, f)$, and that on the training data, i.e., $err(D_n, f)$. To understand this gap, we decompose the generalization error into two terms:

$$
err(P_U, f) - err(D_n, f) = \\
\underbrace{err(P_S, f) - err(D_n, f)}_{\text{IID error}} + \underbrace{err(P_U, f) - err(P_S, f)}_{\text{CG error}}
$$
(5)

The "IID error" can be thought of as the generalization error under the IID assumption. The "CG error" is the focus here.

**2) Function selection.** As for the CG error, even though we eliminate the aforementioned influence by evaluating all functions $f$ on the target distribution $P_U$, there is still an uneliminated influence from the insufficient data. This influence comes from the fact that the function $f$ is sampled from $\mathcal{A}(D_n)$ instead of $\mathcal{A}(P_S^{(T)})$. In short, the learning algorithm operates on the insufficient data. Based on this, we introduce $\kappa_n \triangleq \max_{(P_U, P_E)} \frac{\left| \mathbb{E}_{D_n \sim P_S}[err(P_U, \boldsymbol{f}_{D_n}) - err(P_S, \boldsymbol{f}_{D_n})] \right|}{|err(P_U, \boldsymbol{f}_S) - err(P_S, \boldsymbol{f}_S)|}$ (note that $\boldsymbol{f}_{D_n} \sim \mathcal{A}(D_n)$) to decouple these influences. $\kappa_n$ quantifies the variation in the performance gap between the support distribution and the target distribution over different numbers of training samples. The $\kappa_n$ satisfies that $\lim_{n \to \infty} \kappa_n = 1$. This indicates that the influence of $\kappa_n$ disappears when given infinite data to learn.

**Summary** Based on the above analysis, we can decompose the generalization error as:

$$
|err(P_U, \boldsymbol{f}_{D_n}) - err(D_n, \boldsymbol{f}_{D_n})| \\
\leq |err(P_S, \boldsymbol{f}_{D_n}) - err(D_n, \boldsymbol{f}_{D_n})| \quad (6) \\
+ \kappa_n |err(P_U, \boldsymbol{f}_S) - err(P_S, \boldsymbol{f}_S)|.
$$

As $|err(P_S, \boldsymbol{f}_{D_n}) - err(D_n, \boldsymbol{f}_{D_n})|$ can be an upper bound using any generalization theory with IID assumption,

$|err(P_U, \boldsymbol{f}_S) - err(P_S, \boldsymbol{f}_S)|$ is focused in this paper, as this term comes from the nature of compositional generalization.

## 5.2. Theorem

In this part, we aim to provide a generalization bounded by the statistical properties of the DCG problem. We start with the assumptions used:

**Definition 5.1.** The incompatibility between the composition rule and the learning algorithm is defined as

$$
\begin{aligned}
\tau(T, \mathcal{A}) = \mathbb{E}_{\boldsymbol{P}_E^{(T)} \sim g(T)} \sup_{M_1, M_2 \subset E} \sup_{M_1' \subset M_1, M_2' \subset M_2} \\
\left| err(\boldsymbol{P}_{M_1'}^{(T)}, \boldsymbol{f}_{M_1}) - err(\boldsymbol{P}_{M_2'}^{(T)}, \boldsymbol{f}_{M_2}) \right|,
\end{aligned}
\tag{7}
$$

where $\boldsymbol{f}_{M_1} \sim \mathcal{A}(\boldsymbol{P}_{M_1})$ and $\boldsymbol{f}_{M_2} \sim \mathcal{A}(\boldsymbol{P}_{M_2})$.

**Remark 5.2.** Given a random variable $\boldsymbol{T}$, $\tau(\boldsymbol{T}, \mathcal{A}) = \mathbb{E}_{T \sim P_{\boldsymbol{T}}} \tau(T, \mathcal{A})$.

**Assumption 5.3.** ($L$-bounded) The error function $err(\cdot, \cdot)$ is $L$-bounded, i.e. for all valid inputs $P, f$, we have $|err(P, f)| \leq L$.

**Remark 5.4.** Assumption 5.3 requires that the error is bounded. If the solution performs poorly on a small subset of data points but performs well on the rest, the average error could be disproportionately large due to extreme errors in that small subset without this assumption. This assumption can be easily satisfied by modifying the original error using a $\min(err, \text{bound})$ operation. Alternatively, a bounded error measure, such as 1-accuracy, which ranges between 0 and 1, can be used.

**Assumption 5.5.** Given the distribution $P_E^{(T)} = g(T, \xi)$, there exist functions $m_\xi, m_T$ such that $T = m_T(P_E^T)$ and $\xi = m_\xi(P_S^{(T)})$ for any $S \in E$ and $S \neq \emptyset$.

**Remark 5.6.** Assumption 5.4 requires that the DCG problem is solvable. Our bound does not apply to DCG problems that are entirely unsolvable. This assumption ensures that, given all distributions, we can learn how the given components are combined. For example, if provided with images of various shapes and colors, we should be able to understand how shape and color interact to form specific images, such as a the image of red triangle. This assumption guarantees that there exists a way to recover these compositional rules from the data.

**Theorem 5.7.** *Under Assumption 5.3 and 5.5, given training data $D_n \in \mathcal{Z}^n$ sampled from the support distribution $P_S$, learning algorithm $\mathcal{A}$, then we have*

$$
\begin{aligned}
&\left| \mathbb{E}[err(P_U, \boldsymbol{f}_{D_n}) - err(D_n, \boldsymbol{f}_{D_n})] \right| \\
&\leq GenIID + \Phi_n(I_{\mathcal{A}}(\boldsymbol{f}_S; \boldsymbol{T} | \boldsymbol{P}_S^{(\boldsymbol{T})})) + \tau(\boldsymbol{T}, \mathcal{A}),
\end{aligned}
\tag{8}
$$

*where $GenIID$ denotes any generalization error bound with IID assumption, the subscript of $I_{\mathcal{A}}(\cdot; \cdot)$ denotes influence of $\mathcal{A}$ through $Q^{(\mathcal{A})}$, and $\Phi_n(x) \triangleq$*

$\kappa_n L \sqrt{\min\{x/2, 1 - \exp(-x)\}}$ *where $\lim_{n \to \infty} \kappa_n = 1$. Note that $\Phi_n(x)$ is a monotonically increasing function with respect to $x$.*

## 5.3. Analysis

**1. Knowledge composition.** As shown in the previous analysis, the generalization error of the DCG comes from two sources: insufficient data and knowledge composition. In this paper, we examine our generalization bound when infinite data is given so that we can focus on the knowledge composition. In this situation, we come to the following conclusion:

**Corollary 5.8.** *Under Assumption 5.3 and 5.5, and $\lim_{n \to \infty} GenIID = 0$, then we have*

$$
\begin{aligned}
&|\mathbb{E}[err(P_U, \boldsymbol{f}_S) - err(P_S, \boldsymbol{f}_S)]| \\
&\leq \phi(I_{\mathcal{A}}(\boldsymbol{f}_S; \boldsymbol{T} | \boldsymbol{P}_S^{(\boldsymbol{T})})) + \tau(\boldsymbol{T}, \mathcal{A}),
\end{aligned}
\tag{9}
$$

*where $\phi(x) = L\sqrt{\min\{x/2, 1 - \exp(-x)\}}$.*

**Remark 5.9.** In this corollary, we assume that $\lim_{n \to \infty} GenIID = 0$. Recall that $GenIID$ can be equal to any generalization bound under IID condition. There are many IID generalization bound that can ensure $\lim_{n \to \infty} GenIID \to 0$, including VC dimension (Vapnik & Chervonenkis, 2015), Rademacher complexity (Bartlett & Mendelson, 2002), covering number (Shalev-Shwartz & Ben-David, 2014), algorithm stability (Bousquet & Elisseeff, 2002; Hardt et al., 2016) and information-theoretic analysis (Xu & Raginsky, 2017; Russo & Zou, 2016).

**Remark 5.10.** This corollary reveals a small value of $\phi(I_{\mathcal{A}}(\boldsymbol{f}_S; \boldsymbol{T} | \boldsymbol{P}_S^{(\boldsymbol{T})})) + \tau(\boldsymbol{T}, \mathcal{A})$ is essential for knowledge composition.

**2. Trade-off between the compatibility and MI.** The previous analysis reveal the relation between $\phi(I_{\mathcal{A}}(\boldsymbol{f}_S; \boldsymbol{T} | \boldsymbol{P}_S^{(\boldsymbol{T})})) + \tau(\boldsymbol{T}, \mathcal{A})$ and knowledge composition. However, the relationship between $\phi(I_{\mathcal{A}}(\boldsymbol{f}_S; \boldsymbol{T} | \boldsymbol{P}_S^{(\boldsymbol{T})}))$ and $\tau(\boldsymbol{T}, \mathcal{A})$ is still unclear. To understand this relation, we must first examine the relationships among the elements involved in the calculation of mutual information, as illustrated in the following diagram:

$$
\begin{array}{ccc}
P_S^{(T)} \longrightarrow \boldsymbol{f}_S \longleftarrow \mathcal{A} \\
\searrow \qquad \downarrow \\
T \longrightarrow \tau(T, \mathcal{A})
\end{array}
\tag{10}
$$

From this diagram, we discover that a trade-off exists between $\Phi_n(I_{\mathcal{A}}(\boldsymbol{f}_S; \boldsymbol{T} | \boldsymbol{P}_S^{(\boldsymbol{T})}))$ and $\tau(\boldsymbol{T}, \mathcal{A})$. More specifically, when no constraint is placed on $\tau(\mathcal{A}, T)$, $\boldsymbol{f}_S$ and $T$ are independent such that $I_{\mathcal{A}}(\boldsymbol{f}_S; \boldsymbol{T} | \boldsymbol{P}_S^{(\boldsymbol{T})}) = 0$. In this scenario, the generalization bound can become very large

because $\tau(\mathcal{A}, T)$ can be very large. One possible solution is to impose a constraint such that $\tau(\mathcal{A}, T) \leq \epsilon$ when design the learning algorithm. By applying this constraint, we create a dependency between $\mathcal{T}$ and $\mathcal{A}$ through conditioning on $\tau(\mathcal{A}, T)$, which leads a non-zero value of $I_{\mathcal{A}}(\boldsymbol{f}_S; \boldsymbol{T} | \boldsymbol{P}_S^{(\boldsymbol{T})})$.

**3. Compared with previous studies** 1) **From a technique perspective,** this is the first paper that decouples the influence of finite samples and knowledge composition in generalization analysis in out of distribution. (Netanyahu et al., 2023; Qiu et al., 2021; Oren et al., 2020; Hosseini et al., 2022). This decoupling allows us to focus on the influence of knowledge composition on generalization error, which is at the core of DCG problems. What's more, it allows us to reuse the knowledge from the generalization analysis in the IID situation and reduce the duplication of work. **2) Compared with generalization bounds in DCG,** Previous works (Netanyahu et al., 2023; Dong & Ma, 2022) provide generalization bound methods for DCG problems that are tailored to specific tasks. This means that their works mainly consider the problem using a specific composition rule. The unique features of our theory are that: 1) Our bound is tractable for different composition rules; 2) Our bound connects the generalization behaviour with the mutual information "$I_{\mathcal{A}}(\boldsymbol{f}_S; \boldsymbol{T} | P_S^{(\boldsymbol{T})})$"; this further reveals the statistical mechanism of the compositional generalization. **3) To illustrate the tightness of our bounds,** we compare our bound with that of Ben-David et al. (2010), which is a general bound for out-of-distribution generalization and is therefore comparable to ours. The details are given in the Appendix B.4. Here we list the results of the comparison. We find that we cannot simply say that one method is tighter than the other. We divided the DCG tasks into two types: one dominated by the learning algorithm and one dominated by the function space. In the first situation, the performance of the DCG is highly dependent on the learning algorithm and our bound is much better than Ben-David et al. (2010). This is reasonable because Ben-David et al. (2010) doesn't take into account the influence of the learning algorithm. When it comes to the second situation, we find that our bound is better when there are relatively good support distributions, i.e. when $|S|$ is large. On the other hand, the (Ben-David et al., 2010) is better when $|S|$ is small.

# 6. Experiment

## 6.1. Experiment Design

**1. Components and Compositional rule**: We construct two words set A,B satisfying $|A| = |B| = 1000$ and their corresponding element $a_1, a_2 \subset A$ and $b_1, b_2 \subset B$. $a_1, a_2$ is a partition of $A$ and the same as $b_1, b_2$. $|a_1| = |a_2| = |b_1| = |b_2| = 500$. The composition rule can be any function that satisfy the following form: $(e_1, e_2) \rightarrow e_1 e_2 e_1 e_2 e_1 e_1$.

And we construct 64 composition functions, referred as $T_1, T_2, \cdots, T_{64}$

**2. Distribution Split**: The support distribution takes the elements in the set $\{(e_1, e_2)|(e_1, e_2) \in a_1 \times b_1 \cup a_2 \times b_1 \cup a_1 \times b_2\}$. The target distribution take elements in the set $\{(e_1, e_2)|(e_1, e_2) \subset a_2 \times b_2\}$. It is easy to verify that these designs satisfy the requirement listed in Section 3.

**3. Sequence design**: The input sequence is "$e_1, e_2, r_1, r_2, r_3, \#$", where $r_1, r_2, r_3$ are random words that simulate the randomness. The expected completed sequence is "$e_1, e_2, r_1, r_2, r_3, \#, e_1, e_2, e_1, e_2, e_1, e_1$" if the composition rule is $(e_1, e_2) \rightarrow e_1, e_2, e_1, e_2, e_1, e_1$.

**4. Learning algorithm design**: In our paper, we define the learning algorithm as the mapping between data and the learned function, encompassing a broader concept than just the optimizer. To simulate learning algorithms with varying inductive biases and adaptivity, we adopt the following approach:

1) We employ the GPT-2 model with two configurations:

- Setting 1: 4 layers, 4 attention heads, and an embedding size of 128.

- Setting 2: 6 layers, 8 attention heads, and an embedding size of 256.

2) We pretrain the GPT-2 model using different pretraining data schedules. The pretraining data is generated from a subset of composition rules same to those in the downstream task, but with entirely different words. This setup allows us to create learning algorithms with different inductive biases and adaptivity while preventing data leakage.

## 6.2. Experiemnts on trade-off and non-trade-off improvement

On of the key point in this paper is that the non-trade-off improvement has to rely on the adaptivity of learning algorithm (detail see beyond trade-off page 5). To verify this conclusion, calculate $I_{\mathcal{A}, \beta}(\tilde{T} = T, P_S)$, which is a measure of adaptivity used in out paper, and GACC, which is the average performance across all the tasks with compositional rule in $T_1, T_2, \cdots, T_{64}$. The results are given in Table 1.

## 6.3. Experiments on Generalization Bounds

We conduct the experiments with different rule complexity using the best pretrain setting in previous section. Rule complexity refers to the length of the rule on the output side. For example, the rule complexity of $(e_1, e_2) \rightarrow e_1, e_2, e_1, e_2, e_1, e_1$ is 6, while the rule complexity of $(e_1, e_2) \rightarrow e_1, e_2, e_1, e_2, e_1, e_1, e_1, e_2$ is 8. The results (given in table 2) indicate that our generalization bound

| $I_{\mathcal{A},\beta}(\tilde{T} = T, P_S)$ | 0.073 | 0.115 | 0.125 | 0.281 | 0.362 | 0.462 | 0.481 | 0.505 | 0.527 | 0.564 |
| GACC | 0.605 | 0.591 | 0.565 | 0.633 | 0.696 | 0.750 | 0.772 | 0.789 | 0.752 | 0.776 |

*Table 1.* Values of $I_{\mathcal{A},\beta}(\tilde{T} = T, P_S)$ and GACC over 10 instances

is more tighter than the bound of Ben-David et al.

| **Rule Complexity** | 6 | 8 | 10 | 12 |
| --- | --- | --- | --- | --- |
| CG Error | 0.223 | 0.262 | 0.301 | 0.342 |
| Ben-David et al. | 0.622 | 0.680 | 0.701 | 0.690 |
| Ours | 0.271 | 0.295 | 0.351 | 0.372 |

*Table 2.* Performance comparison across rule complexities

## 7. Discussion

*Q: What are the key properties of data-centric approaches to solving the DCG problem?*

Our theory suggests that a data-centric approach is fundamental for achieving non-trade-off improvements. It highlights the following key properties of data-centric methods:

1. Data-centric methods should effectively leverage information from the data itself. Injecting human task-specific knowledge into method design—such as using specialized model architectures or loss functions—may hinder the method's ability to learn directly from the data.

2. Theory 5.7 further asserts that compatibility between the learning algorithm and the data is crucial. This implies that the learning algorithm should achieve uniform performance across different compositions within the support distribution. For example, if the support distribution includes red triangles and blue rectangles, the model's performance on red triangles and blue rectangles should be similar.

3. Regarding the requirements for the data engineering phase, our theory supports the co-design of both the solution (including network structure and objective function) and data collection. Since the value of $I_{\mathcal{A},\beta}(\tilde{T} = T, P_S)$ in our theory depends on both the learning algorithm and the data, our theory cannot prescribe a universally optimal data development method independent of the specific approach. However, certain data quality requirements, such as the absence of label noise, are absolutely essential.

## 8. Limitation

This paper aims to provide a theoretical understanding of compositional generalization. Consequently, the analysis presented here does not directly address specific DCG problems. However, we argue that the theoretical insights are valuable and can inspire the development of better methods. These findings include:

1) (Section 4) Improving the adaptivity of the proposed method is crucial. Without adaptivity, we run the risk of facing a zero-sum situation, where improving performance on one task may lead to reduced performance on others.

2) (Section 5) We have found that the knowledge composition ability of a method for a given task can be assessed using $\Phi(I_{\mathcal{A}}(\boldsymbol{f}_S; \boldsymbol{T}|\boldsymbol{P}_S^{(\boldsymbol{T})})) + \tau(\boldsymbol{T}, \mathcal{A})$. This reveal the importance of the learning algorithm that not only be compatible with the compositional rule but also be sure its output less influenced by the compositional rule.

In summary, this work offers insights into the generalization mechanisms underlying DCG problems. These insights, along with the introduction of new concepts—particularly the connection between statistical relations and generalization error—can be leveraged to guide the development of new methods.

## 9. Conclusion

This paper aims to understand the generalization mechanisms of the DCG from a statistical perspective. This serves as a complementary view to previous studies. More specifically, our findings include: 1) We propose a new way to model the internal bias and the adaptivity of the learning algorithm separately. Based on this, we propose the $\mu$ measure to analyse the trade-off and non-trade-off improvement. 2) To bridge the statistic properties of the learning algorithm with its knowledge composition capacity, we first provide a way to separates the influence of the insufficiency data and that of the knowledge composition. Then, we identify that small $\Phi(I_{\mathcal{A}}(\boldsymbol{f}_S; \boldsymbol{T}|\boldsymbol{P}_S^{(\boldsymbol{T})})) + \tau(\boldsymbol{T}, \mathcal{A})$ is important for better knowledge composition ability.

### Acknowledgements

The authors gratefully acknowledge the support from the National Natural Science Foundation of China (Grant No. 62088102).

### Impact Statement

The primary goal of this paper is to deepen human understanding of a specific machine learning problem. Our work doesn't have a direct influence on society. However, future works based on our work may influence society but it is unpredictable currently.

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

# A. Proof of Invariant Measure

**Theorem A.1.** *(Invariant Property) There exists at least one function $\alpha$ such that for any $\beta$, the $\mu$-measure satisfies the following conditions:*

- *(1) For any $\boldsymbol{T}$, $\boldsymbol{P}_S^{(T)}$ and $\mathcal{A}$, we have $\mathbb{E}_{\boldsymbol{T}, \boldsymbol{P}_S^{(T)}} \log \alpha_{\mathcal{A},\beta}(\boldsymbol{T}, \boldsymbol{P}_S^{(\boldsymbol{T})}) = I_{\mathcal{A},\beta}(\tilde{\boldsymbol{T}} = \boldsymbol{T}; \boldsymbol{P}_S^{(T)})$;*

- *(2) For any $\mathcal{A}_1, \mathcal{A}_2$, the following equation holds*

$$\mu_\beta(\mathcal{A}_1) = \mu_\beta(\mathcal{A}_2), \tag{11}$$

*where $\mu(\mathcal{A}) = \sum_T \mathbb{E}_{P_S^{(T)} \sim g(T)} \mu(T, P_S^{(T)}, \mathcal{A})$.*

*Proof.* The key to prove this theory is to find a $\alpha$ that satisfies the condition (1) and (2).

- We construct $\alpha_{\mathcal{A},\beta}(T, P_S^{(T)}) = \frac{Q^{(\mathcal{A},\beta)}(P_S^{(T)}|\tilde{T}=T)}{Q^{(\mathcal{A},\beta)}(P_S^{(T)})}$.

- In Proposition A.2, we prove that the $\alpha$ satisfies the condition (1)

- In Proposition A.3, we prove that the $\alpha$ satisfies the condition (2).

Combine the results above, the theorem is proved.

$\square$

**Proposition A.2.** *For any learning algorithm $\mathcal{A}$, given two random variable $\boldsymbol{T}, \boldsymbol{P}_S^{(T)}$, we have*

$$\mathbb{E}_{\boldsymbol{T}, \boldsymbol{P}_S^{(T)}} \log \alpha_{\mathcal{A},\beta}(\boldsymbol{T}, \boldsymbol{P}_S^{(\boldsymbol{T})}) = I_{\mathcal{A},\beta}(\tilde{\boldsymbol{T}} = \boldsymbol{T}; \boldsymbol{P}_S^{(T)}). \tag{12}$$

*Proof.* According to the equation $\alpha_{\mathcal{A},\beta}(T, P_S^{(T)}) = \frac{Q^{(\mathcal{A},\beta)}(P_S^{(T)}|\tilde{T}=T)}{Q^{(\mathcal{A},\beta)}(P_S^{(T)})}$, we have

$$
\begin{aligned}
&\mathbb{E}_{\boldsymbol{T}, \boldsymbol{P}_S^{(T)}} \log \alpha_{\boldsymbol{T}}(\mathcal{A}, \boldsymbol{P}_S^{(\boldsymbol{T})}) \\
&= \sum_{T, P_S^{(T)}} Q^{(\mathcal{A},\beta)}(\tilde{T} = T, P_S^{(T)}) \log \frac{Q^{(\mathcal{A},\beta)}(P_S^{(T)}|\tilde{T} = T)}{Q^{(\mathcal{A},\beta)}(P_S^{(T)})} \\
&= \sum_{T, P_S^{(T)}} Q^{(\mathcal{A},\beta)}(\tilde{T} = T, P_S^{(T)}) \log \frac{Q^{(\mathcal{A},\beta)}(P_S^{(T)}, \tilde{T} = T)}{Q^{(\mathcal{A},\beta)}(P_S^{(T)}), Q^{(\mathcal{A},\beta)}(T)} \\
&\overset{(\star)}{=} I_{\mathcal{A},\beta}(\tilde{\boldsymbol{T}} = \boldsymbol{T}, \boldsymbol{P}_S^{(\tilde{T}=T)}),
\end{aligned}
\tag{13}
$$

where $(\star)$ is due to the definition of the mutual information. Therefore, the Proposition is established. $\square$

**Proposition A.3.** *For any $\mathcal{A}_1, \mathcal{A}_2$ and $\beta$, we have*

$$\mu_\beta(\mathcal{A}_1) = \mu_\beta(\mathcal{A}_2), \tag{14}$$

*where $\mu(\mathcal{A}) = \sum_T \mathbb{E}_{P_S^{(T)} \sim g(T)} \mu(T, P_S^{(T)}, \mathcal{A})$.*

*Proof.* According to the bayes rule, we have

$$Q(\tilde{T} = T|P_S^{(T)}) = \frac{Q(P_S^{(T)}|\tilde{T} = T)Q(\tilde{T} = T)}{Q(P_S^{(T)})}. \tag{15}$$

Since $\tilde{T}$ is the prediction, therefore $T$ is available when $\tilde{T} = T$. Based on this, we have

$$Q^{(\mathcal{A},\beta)}(P_S^{(T)}|\tilde{T} = T) = Q^{(\mathcal{A},\beta)}(P_S^{(\tilde{T})}|\tilde{T} = T). \tag{16}$$

According to the definition of $\alpha_{\mathcal{A},\beta}(T, P_S^{(T)})$, we have

$$\alpha_{\mathcal{A},\beta}(T, P_S^{(T)}) = \frac{Q(\tilde{T} = T|P_S)}{Q(\tilde{T} = T)}. \tag{17}$$

First, we consider the learning algorithms that $\alpha_{\mathcal{A},\beta}(T, P_S^{(T)}) \neq 0$ for all $T, P_S$ According to the definition of $\mu(\cdot)$, we have

$$
\begin{aligned}
&\mathbb{E}_{P_S^{(T)} \sim g(T)} \mu_\beta(T, P_S, \mathcal{A}) \\
&= \mathbb{E}_{P_S^{(T)} \sim g(T)} \frac{Q^{(\mathcal{A},\beta)}(\tilde{T} = T|P_S)}{\alpha_{\mathcal{A},\beta}(T, P_S^{(T)})} \\
&= \mathbb{E}_{P_S^{(T)} \sim g(T)} Q^{(\mathcal{A},\beta)}(\tilde{T} = T) \\
&= \sum_{P_S^{(T)}} Q^{(\mathcal{A},\beta)}(P_S^{(T)}) Q^{(\mathcal{A},\beta)}(\tilde{T} = T) \\
&= Q^{(\mathcal{A},\beta)}(\tilde{T} = T).
\end{aligned}
\tag{18}
$$

For all $\mathcal{A}$, we have

$$\sum_T Q^{(\mathcal{A},\beta)}(\tilde{T} = T) = \sum_{\tilde{T}} Q^{(\mathcal{A},\beta)}(\tilde{T}) = 1. \tag{19}$$

Combined all the equation above, we have

$$
\begin{aligned}
&\sum_T \mathbb{E}_{P_S^{(T)} \sim g(T)} \mu_\beta(T, P_S, \mathcal{A}_1) \\
&= \sum_T \mathbb{E}_{P_S^{(T)} \sim g(T)} Q^{(\mathcal{A}_1,\beta)}(\tilde{T} = T) \\
&= \sum_T \mathbb{E}_{P_S^{(T)} \sim g(T)} Q^{(\mathcal{A}_2,\beta)}(\tilde{T} = T) \\
&= \sum_T \mathbb{E}_{P_S^{(T)} \sim g(T)} \mu_\beta(T, P_S, \mathcal{A}_2).
\end{aligned}
\tag{20}
$$

Given a learning algorithm $\mathcal{A}_1$ there exists a set $\mathcal{V}$, such that for all $(P_S, T \in \mathcal{V})$, we have $\alpha_T(\mathcal{A}_1, P_S^{(T)}) = 0$. The learning algorithm $\mathcal{A}_2$ satisties that $\alpha_T(\mathcal{A}_1, P_S^{(T)}) > 0$ for all $T, P_S$. If this theorem holds, we expect that

$$
\begin{aligned}
&\sum_T \mathbb{E}_{P_S^{(T)} \sim g(T)} \mu_\beta(T, P_S, \mathcal{A}_2) \\
&= \sum_T \mathbb{E}_{P_S^{(T)} \sim g(T)} \mu_\beta(T, P_S, \mathcal{A}_1) \\
&= \sum_T \mathbb{E}_{P_S^{(T)} \sim g(T)} \mu_\beta(T, P_S, \mathcal{A}_1) \mathbf{1}[(T, P_S) \notin \mathcal{V}] \\
&+ \sum_T \mathbb{E}_{P_S^{(T)} \sim g(T)} \mu_\beta(T, P_S, \mathcal{A}_1) \mathbf{1}[(T, P_S) \in \mathcal{V}].
\end{aligned}
\tag{21}
$$

Because $\mu_\beta(T, P_S, \mathcal{A}_1) > 0$ holds for all inputs, we have

$$\sum_T \mathbb{E}_{P_S^{(T)} \sim g(T)} \mu_\beta(T, P_S, \mathcal{A}_1) \mathbf{1}[(T, P_S) \notin \mathcal{V}] \geq 0, \tag{22}$$

$$\sum_T \mathbb{E}_{P_S^{(T)} \sim g(T)} \mu_\beta(T, P_S, \mathcal{A}_1) \mathbf{1}[(T, P_S) \in \mathcal{V}] \geq 0. \tag{23}$$

Obviously, we have

$$
\begin{aligned}
&\sum_T \mathbb{E}_{P_S^{(T)} \sim g(T)} \mu_\beta(T, P_S, \mathcal{A}_1) \mathbf{1}[(T, P_S) \notin \mathcal{V}] \\
&\leq \sum_T \mathbb{E}_{P_S^{(T)} \sim g(T)} \mu_\beta(T, P_S, \mathcal{A}_1) = 1.
\end{aligned}
\tag{24}
$$

Therefore, we can find a value assignment that assign the value between 0 and 1 to $\mu_\beta(T, P_S, \mathcal{A}_1)$ for all $(P_S, T) \in \mathcal{V}$ such that the Theorem holds. $\qquad\square$

## B. Proof of Generalization Bound

### B.1. Preliminary: Definition and useful lemma

In the following, we give the measure for the distribution, i.e. Wasserstein Distance and the some common used function assumption, i.e. Lipschitz assumption and homeomorphis assumption.

**Definition B.1.** (Wasserstein Distance). For any $p \geq 1$, the $p$-Wasserstein distance between two pobability measures $P, \mathbb{Q}$ on the space $\mathcal{W}$ with metric $d_\mathcal{W}$ is defined as:

$$
\mathbb{W}_p(P, \mathbb{Q}) = \inf_{M \in \Gamma(P, \mathbb{Q})} (\mathbb{E}_{(W,W') \sim M}[d_\mathcal{W}^p(W, W')])^{1/p},
\tag{25}
$$

where $\Gamma(P, \mathbb{Q})$ denotes the collection of all measures on $W \times W$ with the marginals $P$ and $\mathbb{Q}$ on the first and second components respectively.

**Definition B.2.** (Lipschitz) Given two metric spaces $(\mathcal{M}, d_\mathcal{M})$ and $(\mathcal{N}, d_\mathcal{N})$, where $d_\mathcal{M}$ and $d_\mathcal{N}$ denote the metrics on $\mathcal{M}$ and $\mathcal{N}$. A function $h : \mathcal{M} \to \mathcal{N}$ is $L$-Lipschitz if for all $m_1, m_2 \in \mathcal{M}$, we have $d_\mathcal{N}(h(m_1), h(m_2)) \leq L d_\mathcal{M}(m_1, m_2)$.

**Lipschitz assumption is commonly used assumption** The majority of research relies on the Lipschitz assumption when analyzing generalization behavior. Some studies attempt to alleviate this assumption by substituting it with its weaker counterpart. However, as the primary focus of this paper does not lie in removing the Lipschitz assumption, we defer this task to future work.

**Definition B.3.** (homeomorphism) A continuous function $f$ is called a homeomorphism if it is a bijection function and its inverse function $f^{-1}$ is continuous as well.

**Definition B.4.** (Total Variation) The total variation between two probability distributions $P$ and $\mathbb{Q}$ on $\mathcal{W}$ is

$$
\text{TV}(P, \mathbb{Q}) \triangleq \sup_{A \in \mathcal{W}} \{P(A) - \mathbb{Q}(A)\}
\tag{26}
$$

**Definition B.5.** (Discrete Metric) The discrete metric is $d(x, y) \triangleq \mathbf{1}[x \neq y]$, where $\mathbf{1}$ is the indicator function.

**Lemma B.6.** *(Rademacher Complexity (from Mohri et al. (2018))) Let $\mathcal{F}$ be a family of functions. Given a distribution $P$ and a samples $D_n = \{z_1, \cdots, z_n\} \sim P^{\otimes n}$, the following holds for all $g \in \mathcal{F}$:*

$$
\mathbb{E}_{D_n \sim P^{\otimes n}}[err(P, f) - err(D_n, f)] \leq 2\mathcal{R}_n(\mathcal{F}),
\tag{27}
$$

*where $\mathcal{R}_n = \mathbb{E}_{\sigma, D_n}[\sup_{f \in \mathcal{F}} \frac{1}{n} \sum_{i=1}^n \sigma_i f(x_i)]$ with $\sigma_i$ being independent uniform random variables taking values in $\{-1, +1\}$.*

**Lemma B.7.** *For two pobability measures $P, \mathbb{Q}$ on the space $\mathcal{W}$ with metric $d_\mathcal{W}$, the $1$-Wasserstein distance between $P$ and $\mathbb{Q}$ can be represented as:*

$$
\mathbb{W}_1(P, \mathbb{Q}) = \frac{1}{L} \sup_{h \in \mathcal{H}} \mathbb{E}_{w \sim P} h(w) - \mathbb{E}_{w \sim \mathbb{Q}} h(w),
\tag{28}
$$

*where $\mathcal{H}$ denotes the function spaces containing function with Lipschitz constant less or equal to $L$.*

## B.2. Proof of Theorem

To prove this theorem, we first start with a important lemma:

**Lemma B.8.** *Under Assumption 5.3, given training data $D_n \in \mathcal{Z}^n$ sampled from the support distribution $P_S$, learning algorithm $\mathcal{A}$, then we have*

$$|\mathbb{E}[err(P_U, f) - err(D_n, f)]| \leq GenIID + \kappa_n L \mathbb{W}_1(Q_{\boldsymbol{f}_S}^{(\mathcal{A})}, Q_{\boldsymbol{f}_E}^{(\mathcal{A})}) + \tau(\boldsymbol{T}, \mathcal{A}), \tag{29}$$

*where $GenIID$ denotes any generalization error bound with IID assumption, $\Phi(x) \triangleq \sqrt{\min\{x/2, 1 - \exp(-x)\}}$, and $\kappa_n \triangleq \max \frac{|\mathbb{E}_{D_n \sim P_S}[err(P_U, \boldsymbol{f}_{D_n}) - err(P_S, \boldsymbol{f}_{D_n})]|}{|err(P_U, \boldsymbol{f}_S) - err(P_S, \boldsymbol{f}_S)|}$ (note that $\boldsymbol{f}_{D_n} \sim \mathcal{A}(D_n)$).*

*Proof.* We can decomposite $err(P_U, f) - err(D_n, f)$ as

$$\mathbb{E}[err(P_U, f) - err(D_n, f)] = \underbrace{\mathbb{E}[err(P_S, \boldsymbol{f}_{D_n}) - err(D_n, \boldsymbol{f}_{D_n})]}_{(1)} + \underbrace{\mathbb{E}[err(P_U, \boldsymbol{f}_{D_n}) - err(P_S, \boldsymbol{f}_{D_n})]}_{(2)}. \tag{30}$$

Because $(1)$ is the generalization bound in IID situation, we can upperbound it with any IID bound. Therfore, we can bound ”$(1)$” term with $GenIID$ to denotes any upper bound of IID. Then, we only need to focus on the $(2)$ term, which is the essential part of DCG.

Then, we have

$$
\begin{aligned}
&\left| err(P_U^{(T)}, \boldsymbol{f}_S) - err(P_S^{(T)}, \boldsymbol{f}_S) \right| \\
&= \left| err(P_U^{(T)}, \boldsymbol{f}_S) - err(P_U^{(T)}, \boldsymbol{f}_E) + err(P_U^{(T)}, \boldsymbol{f}_E) - err(P_S^{(T)}, \boldsymbol{f}_S) \right| \\
&\leq \left| err(P_U^{(T)}, \boldsymbol{f}_S) - err(P_U^{(T)}, \boldsymbol{f}_E) \right| + \left| err(P_U^{(T)}, \boldsymbol{f}_E) - err(P_S^{(T)}, \boldsymbol{f}_S) \right| \\
&\leq \left| err(P_U^{(T)}, \boldsymbol{f}_S) - err(P_U^{(T)}, \boldsymbol{f}_E) \right| + \tau(T, \mathcal{A})
\end{aligned}
\tag{31}
$$

According to Lemma B.7, we have

$$\mathbb{W}_1(P, \mathbb{Q}) = \frac{1}{L} \sup_{h \in \mathcal{H}} \mathbb{E}_{w \sim P} h(w) - \mathbb{E}_{w \sim \mathbb{Q}} h(w). \tag{32}$$

By replacing $h$ in Equation 32 with $err(P_U, \cdot)$, $P$ in Equation 32 with $Q_{\boldsymbol{f}_S}^{(\mathcal{A})}$ and $\mathbb{Q}$ in Equation 32 with $P_{\mathcal{F}_c}$ we obtain that

$$
\begin{aligned}
&\mathbb{W}_1(Q_{\boldsymbol{f}_S}^{(\mathcal{A})}, Q_{\boldsymbol{f}_E}^{(\mathcal{A})}) \\
&\geq \frac{1}{L}\left( \mathbb{E}_{f \sim Q_{\boldsymbol{f}_S}^{(\mathcal{A})}} err(P_U, f) - \mathbb{E}_{f \sim Q_{\boldsymbol{f}_E}^{(\mathcal{A})}} err(P_U, f) \right) \\
&= \frac{1}{L}\left( err(P_U, \boldsymbol{f}_S) - err(P_U, \boldsymbol{f}_E) \right).
\end{aligned}
\tag{33}
$$

By rearranging the equation, we obtain that

$$err(P_U, \boldsymbol{f}_S) - err(P_U, \boldsymbol{f}_E) \leq L \mathbb{W}_1(Q_{\boldsymbol{f}_S}^{(\mathcal{A})}, Q_{\boldsymbol{f}_E}^{(\mathcal{A})}). \tag{34}$$

According to the Definition of $\kappa_n$, we have

$$err(P_U, \boldsymbol{f}_{D_n}) - err(P_S, \boldsymbol{f}_{D_n}) \leq \kappa_n(err(P_U, \boldsymbol{f}_S) - err(P_S, \boldsymbol{f}_S)) \tag{35}$$

Combining the equations above, the result is established. $\qquad \square$

**Theorem B.9.** *Under Assumption 5.3 and 5.5, given training data $D_n \in \mathcal{Z}^n$ sampled from the support distribution $P_S$, learning algorithm $\mathcal{A}$, then we have*

$$\left| \mathbb{E}[err(P_U, \boldsymbol{f}_{D_n}) - err(D_n, \boldsymbol{f}_{D_n})] \right| \leq GenIID + \Phi_n(I_{\mathcal{A}}(\boldsymbol{f}_S; \boldsymbol{T} | \boldsymbol{P}_S^{(\boldsymbol{T})})) + \tau(\boldsymbol{T}, \mathcal{A}), \tag{36}$$

*where $GenIID$ denotes any generalization error bound under IID assumption, $\Phi_n(x) \triangleq \kappa_n L \sqrt{\min\{x/2, 1 - \exp(-x)\}}$, and $\kappa_n \triangleq \max \frac{|\mathbb{E}_{D_n \sim P_S}[err(P_U, \boldsymbol{f}_{D_n}) - err(P_S, \boldsymbol{f}_{D_n})]|}{|err(P_U, \boldsymbol{f}_S) - err(P_S, \boldsymbol{f}_S)|}$ (note that $\boldsymbol{f}_{D_n} \sim \mathcal{A}(D_n)$).*

*Proof.* Start from Lemma B.8, we set the metric between the function space, i.e. $d_{\mathcal{F}}$, as the discrete metric as defined in Definition B.5. Based on this metric, because the $err(\cdot)$ is $L$-bounded, we have for any distribution $\mathbb{Q}$ and $f_1, f_2 \in \mathcal{F}$, $\frac{|err(\mathbb{Q}, f_1) - err(\mathbb{Q}, f_2)|}{d_{\mathcal{F}}(f_1, f_2)} \leq \frac{|L - 0|}{1} = L$, i.e. the $err(\cdot)$ is $L$-Lipschitz.

Then, we can bound $\mathbb{W}_1(Q_{\boldsymbol{f}_S}^{(\mathcal{A})}, Q_{\boldsymbol{f}_E}^{(\mathcal{A})})$ in Lemma B.8 with $\Phi(I_{\mathcal{A}}(\boldsymbol{f}_S; \boldsymbol{T}|P_S^{(\boldsymbol{T})}))$:

$$\mathbb{W}_1(Q_{\boldsymbol{f}_S}^{(\mathcal{A})}, Q_{\boldsymbol{f}_E}^{(\mathcal{A})}) = \mathbb{W}_1(Q_{\boldsymbol{f}_E}^{(\mathcal{A})}, Q_{\boldsymbol{f}_S}^{(\mathcal{A})}) \overset{(\clubsuit)}{=} \text{TV}(Q_{\boldsymbol{f}_E}^{(\mathcal{A})}, Q_{\boldsymbol{f}_S}^{(\mathcal{A})}) \overset{(\heartsuit)}{\leq} \Phi(KL(Q_{\boldsymbol{f}_E}^{(\mathcal{A})}, Q_{\boldsymbol{f}_S}^{(\mathcal{A})})), \tag{37}$$

where $(\clubsuit)$ is due the Theorem 6.15 of Villani et al. (2009),$(\heartsuit)$ is due to the statement in Theorem 6.5 of Polyanskiy & Wu (2014) and Lemma 2 of Rodríguez Gálvez et al. (2021). With some misuses, we denote $Q_{\boldsymbol{f}_S}^{(\mathcal{A})}$ as $Q_{\boldsymbol{f}}^{(\mathcal{A})}|P_S$, where $|$ denotes the condition and the same for $Q_{\boldsymbol{f}_E}^{(\mathcal{A})}$ and $P_{\boldsymbol{f}_U}$. Then, we have

$$\begin{aligned}
\text{KL}(Q_{\boldsymbol{f}_E}^{(\mathcal{A})}, Q_{\boldsymbol{f}_S}^{(\mathcal{A})}) &= \text{KL}([Q_{\boldsymbol{f}}^{(\mathcal{A})}|P_E], [Q_{\boldsymbol{f}}^{(\mathcal{A})}|P_S]) \\
&= \text{KL}([Q_{\boldsymbol{f}}^{(\mathcal{A})}|(P_S, P_U)], [Q_{\boldsymbol{f}}^{(\mathcal{A})}|P_S]) \\
&= \text{KL}([Q_{\boldsymbol{f}}^{(\mathcal{A})}|P_U], Q_{\boldsymbol{f}}^{(\mathcal{A})}|P_S) \\
&= I_{\mathcal{A}}(\boldsymbol{f}; \boldsymbol{P}_U|\boldsymbol{P}_S)
\end{aligned} \tag{38}$$

The notation $[Q_{\boldsymbol{f}}^{(\mathcal{A})}|P_U]$ indicates that the condition $P_U$ only take effect on the distribution $Q_{\boldsymbol{f}}^{(\mathcal{A})}$. While the $KL(\cdot, \cdot|P_S)$ indicates that the condition $P_S$ take effect on all the distributions.

According to the Assumption 5.5, there exists a bijection function between $T$ and $P_U$ when $P_S$ is given. Based on this, we have

$$I_{\mathcal{A}}(\boldsymbol{f}; \boldsymbol{P}_U|\boldsymbol{P}_S) = I_{\mathcal{A}}(\boldsymbol{f}; \boldsymbol{T}|\boldsymbol{P}_S) \tag{39}$$

Combine the equations above, the Theorem is established. $\qquad\square$

## B.3. Proof of Corollary

**Corollary B.10.** *Under Assumption 5.3 and 5.5, and* $\lim_{n \to \infty} GenIID = 0$*, then we have*

$$|\mathbb{E}[err(P_U, \boldsymbol{f}_S) - err(P_S, \boldsymbol{f}_S)]| \leq \Phi(I_{\mathcal{A}}(\boldsymbol{f}_S; \boldsymbol{T}|\boldsymbol{P}_S^{(\boldsymbol{T})})) + \tau(\boldsymbol{T}, \mathcal{A}). \tag{40}$$

*Proof.* Recall that the generalization bound in Theorem 5.7, that

$$\left|\mathbb{E}[err(P_U, \boldsymbol{f}_{D_n}) - err(D_n, \boldsymbol{f}_{D_n})]\right| \leq GenIID + \Phi_n(I_{\mathcal{A}}(\boldsymbol{f}_S; \boldsymbol{T}|\boldsymbol{P}_S^{(\boldsymbol{T})})) + \tau(\boldsymbol{T}, \mathcal{A}), \tag{41}$$

Taking $n \to \infty$, we have

$$\begin{aligned}
&\lim_{n \to \infty} |\mathbb{E}[err(P_U, f) - err(D_n, f)]| \\
&= |\mathbb{E}[err(P_U, \boldsymbol{f}_S) - err(P_S, \boldsymbol{f}_S)]| \\
&= \lim_{n \to \infty} (GenIID + \Phi_n(I_{\mathcal{A}}(\boldsymbol{f}_S; \boldsymbol{T}|P_S^{(\boldsymbol{T})})) + \tau(\boldsymbol{T}, \mathcal{A})) \\
&\overset{(\star)}{=} \lim_{n \to \infty} \Phi_n(I_{\mathcal{A}}(\boldsymbol{f}_S; \boldsymbol{T}|P_S^{(\boldsymbol{T})})) + \tau(\boldsymbol{T}, \mathcal{A}) \\
&= \Phi(I_{\mathcal{A}}(\boldsymbol{f}_S; \boldsymbol{T}|P_S^{(\boldsymbol{T})}) + \tau(\boldsymbol{T}, \mathcal{A}),
\end{aligned} \tag{42}$$

where $(\star)$ is due to the condition $\lim_{n \to \infty} GenIID = 0$ and $\lim_{n \to \infty} \kappa_n = 1$. $\qquad\square$

## B.4. Tightness

Our ability to assert whether our bound is tighter or looser than previous bounds is contingent upon considering the nuanced intricacies of the problems at hand. According to whether the problem is more influenced by the design of learning algorithm or the function space. We have delineated the issue into two distinct categories

*Figure 1.* **Generalization bounds on the toy problem.** The example 1 considers the case where the function space has some bias while the learning algorithm has no bias. The example 2 consider the learning algorithm has certain bias while the function space is powerful to fit data. We find that 1) our bounds can capture the decrease of generalization error in example 1 and 2) our can align with the generalization error in example 2.

- **Function space dominated problem.** In this problem, we posit that the learning algorithm randomly selects functions that minimize loss on the training data, within a function space tailored specifically for the problem at hand.

- **learning algorithm dominated problem.** Here, we assume that the function space is pwerful enough to accommodate any distribution, while the learning algorithm is inclined to favor certain functions over others, provided they minimize loss on the training data.

To illustrate these components, we provide two examples: Example 1 exemplifies a scenario where the function space dominates, whereas Example 2 exemplifies a scenario where the learning algorithm holds sway. In summation, our analysis yields the following conclusion: *We ascertain that our bound achieves greater tightness in the context of the learning algorithm dominated problem. Conversely, in the scenario where the function space dominates, our bound achieves greater tightness solely when provided with an extensive array of supporting distributions.*

**Example setting** We consider thet $|A| = |B| = 10$. The two examples are explored. **Example 1** The function space $\mathcal{F}$ has the properties that 1) For each function $f \in \mathcal{F}$, we have $\sum_{a,b} \mathbb{I}_{err(P_{a,b},f)} = 10$ or $\sum_{a,b} \mathbb{I}_{err(P_{a,b},f)} = 0$. 2) $\forall a \in A, b \in B, err(P_{a,b}, f) = 1$ or $err(P_{a,b}, f) = 0$. The learning algorithm satisfies that $\forall (a, b) \in S$ and for all $f \in \operatorname{supp} Q^{(\mathcal{A})}_{\boldsymbol{f}_S}$ and for all $(a, b) \in S$, we have $err(P_{a,b}, f) = 0$. **Example 2** For all $f \in \operatorname{supp} Q^{(\mathcal{A})}_{\boldsymbol{f}_S}$, for all $(a, b) \in S$, we have $err(P_{a,b}, f) = 0$ and for all $(a, b) \notin S$, we have $err(P_{a,b}, f) = 0$ with probability $c_{a,b} \frac{|S|}{|E|}$ else $err(P_{a,b}, f) = 1$, where $c_{a,b}$ is a ramdonly assigned value for each $(a, b)$ and it takes value between 0.8 and 1. We choose the distance measure $d_{\mathcal{F}}(f_1, f_2) = \sup_{(a,b) \in E} \mathbb{E}(|err(P_{a,b}, f_1) - err(P_{a,b}, f_2)|)$.

**Remark B.11.** In **Example 1**, we delve into the bias stemming from the function space. Here, the function space is relatively constrained, containing only a limited set of functions, including the correct one that attains zero loss. The learning algorithm uniformly selects a function only if it achieves minimal loss on the support distribution. Consequently, the learning algorithm exhibits no inherent bias towards specific functions as long as they achieve minimal loss on the support distributions. In **Example 2**, we explore the bias inherent in the learning algorithm. In this instance, the function space is expansive, encompassing all possible outputs. However, the learning algorithm may assign varying probabilities to functions that achieve zero loss on the support distribution.

We revisit the bounds here:

**1) The results of Ben-David et al. (2010).** Ben-David et al. (2010) has the following conclusion that:

$$err(P_U, \boldsymbol{f}_S) - err(P_S, \boldsymbol{f}_S) \leq d_{\mathcal{F}\Delta\mathcal{F}}(P_U, P_S) + \tau(\boldsymbol{T}, \mathcal{A}), \tag{43}$$

where the $d_{\mathcal{F}\Delta\mathcal{F}}$ is defined as $d_{\mathcal{F}\Delta\mathcal{F}} \triangleq 2 \sup_{f,f' \in \mathcal{F}} |\mathbb{E}_{z \sim P_U}[f(x) \neq f'(x)] - \mathbb{E}_{z \sim P_S}[f(x) \neq f'(x)]|$.

**2) Ours.** The Corollary 5.8 in our work indicates that

$$err(P_U, \boldsymbol{f}_S) - err(P_S, \boldsymbol{f}_S) \leq \Phi(I_{\mathcal{A}}(\boldsymbol{f}_S; \boldsymbol{T} | \boldsymbol{P}^{(\boldsymbol{T})}_S)) + \tau(\boldsymbol{T}, \mathcal{A}). \tag{44}$$

**Results** We compute the generalization bounds and error depicted in Fig. 1, revealing two key observations: Our bound effectively incorporates the impact of the support distribution. In **Example 1**, our generalization bound accurately reflects

the decreasing trend of the generalization error. Similarly, in **Example 2**, our bound aligns with the generalization trends across various support distributions. Our bound accounts for the influence of the learning algorithm. Notably, in **Example 2**, the approach proposed by Ben-David et al. (2010) fails to capture the dynamics accurately. This failure can be attributed to its predominant focus on the function space influence, whereas our analysis recognizes the dominance of the learning algorithm's influence in this example.

