# OpenReview forum: "On the Statistical Mechanisms of Distributional Compositional Generalization"
_ICML.cc/2025/Conference — ICML 2025 poster_

### Official Review · Reviewer_PSGU · 2025-03-14

**Overall Recommendation:** 3

**Summary:**

The paper investigates the statistical mechanisms underlying Distributional Compositional Generalization (DCG), focusing on two key questions: 1) whether methods for one DCG task generalize to others, and 2) what statistical properties determine a learning algorithm’s compositional ability. The authors propose an invariant measure (μ) to unify diverse DCG methods, highlighting data adaptivity as critical for non-trade-off improvements. They derive a generalization bound decoupling IID error and compositional error, linking compositional capacity to mutual information and compatibility between learning algorithms and composition rules. The analysis emphasizes the role of statistical dependencies and algorithmic sensitivity in DCG, offering theoretical insights complementary to prior work.

**Claims And Evidence:**

The claims are supported by theoretical derivations but lack empirical validation.

**Essential References Not Discussed:**

N/A

**Experimental Designs Or Analyses:**

No experiments are provided to validate the theoretical claims. The analysis remains purely mathematical, leaving open questions about the practical relevance of the proposed measures in real-world DCG tasks.

**Methods And Evaluation Criteria:**

The theoretical framework is logically structured, using mutual information and compatibility measures to analyze DCG. However, the absence of empirical benchmarks or synthetic experiments weakens validation. The reliance on abstract statistical formulations may limit applicability to specific DCG tasks without further domain-specific adjustments.

**Other Comments Or Suggestions:**

1. Terms like "knowledge composition" need clearer definitions.
2. The organizational structure of the paper needs adjustment, as readers without theoretical background find it difficult to follow.

**Other Strengths And Weaknesses:**

Strengths include a novel statistical perspective on DCG and the invariant measure unifying diverse methods. Weaknesses include overly abstract formulations and lack of empirical grounding. Clarity suffers from dense notation and undefined terms.

**Questions For Authors:**

1. How might the μ-measure be empirically estimated or validated in practice? Could this inform method selection for DCG tasks?
2. Assumption 5.4 requires recovering \(T\) from \(P_E^{(T)}\). Is this feasible for complex composition rules (e.g., in NLP or robotics)?
3. Are there plans to test the theory on synthetic or real-world DCG benchmarks (e.g., SCAN, COGS)?
4. Could the "composition rule" \(T\) be explicitly defined for canonical DCG tasks (e.g., attribute recombination)?

**Relation To Broader Scientific Literature:**

The work extends statistical learning theory to DCG, contrasting with IID-based generalization frameworks. It connects to NFL theorems by addressing task trade-offs but focuses specifically on compositional rules. Comparisons to Ben-David et al.’s domain adaptation bounds highlight novel aspects (algorithmic compatibility), though empirical validation is missing.

**Theoretical Claims:**

The proofs for Theorem 4.7 and Theorem 5.5 rely on assumptions (e.g., L-bounded errors, composition rule recoverability in Assumption 5.4) that are plausible but not rigorously justified. The mutual information term \(I_{\mathcal{A}}(f_S; T|P_S^{(T)})\) is central but lacks intuitive interpretation in practical settings.

---

> ### Author Rebuttal · Authors · 2025-03-30
>
> > W1: lack empirical validation.
>
> We present new experimental results to validate the theoretical derivations (**see the "Experiments" section in the Rebuttal of Reviewer XpR4 for details**). Below is a summary of our findings:
> 1. We confirm that non-trade-off improvements are strongly correlated with adaptivity of learning algorithms.
> 2. The results demonstrate that our bound is tighter than previous methods, supporting the importance of incorporating information about the learning algorithm and data for tighter bound.
>
> > W2: The reliance on abstract statistical formulations may limit applicability.
>
> The goal of this paper is to offer a broader perspective on DCG tasks. We fully acknowledge the importance of applying the framework to specific DCG tasks; however, substantial efforts have already been dedicated to this aspect in previous works. Rather than replicating those efforts, our paper seeks to address a gap that has not yet been fully explored. The value of our work should not be assessed in isolation. When considered within the broader context of the research community—where numerous theoretical studies on specific tasks already exist—our work provides a complementary perspective that deepens the overall understanding of DCG tasks.
>
> > W3: Terms like "knowledge composition" need clearer definitions.
>
> Knowledge composition refers to a learning algorithm's ability to understand and integrate individual components. Specifically, it reflects the algorithm's performance on DCG tasks when the impact of limited data is eliminated (i.e., given an infinite amount of data). To avoid ambiguity, we plan to replace "knowledge composition" with "component composition" and provide a clearer explanation.   **If there are other terms that require clarification, we would greatly appreciate your feedback and immedately address them.**
>
> > W4: readers without theoretical background find it difficult to follow
>
> The more examples and background will be provided. The examples will cover the core concepts and the assumptions. The background will be listed in the Appendix to cover the basic knowledge of the statistic machine learning. **The detail can be seen in the Section "More examples and background" of Rebuttal to Reviewer nkeW**.  We hope that these effort can make the non-theoretical background reader more easy to read.
>
> > Q1: Estimate of $\mu$ or validated in practice? Could this inform method selection for DCG tasks?
>
> The core value of $\mu$ measure is to provide the analysis of the trade-off and non-trade-off improvement. We design a synthetic task for verify the trade and non-trade-off improvement (See **W1**).
>
> This has the following indication for the method selection for DCG:
> 1) To improvement the learning algorithm in non-trade-off way, it is counterproductive to impose inductive biases or constraints, such as the group constraint, as this would cause the model to loss the data adaptivity ability. A more effective approach is to design a model that can fully leverage the information in the data.
>
> 2) Co-design of the learning algorithm and data engeering is importance, as the non-trade-off improvement replies on a large value of $I_{\mathcal{A},\beta}(\tilde{T}=T,P_S)$. The compatibility between the data the learning algorithm is essential for generalization.
>
> More detail information can be seen in **Q1 in the response to Reviewer nkeW**
>
> > Q2: Clarify Assumptions ?
>
> Assumption 5.3 requires that the error is bounded. For example, the 1-accuracy, which ranges between 0 and 1, can satisfy the assumption.
>
> Assumption 5.4 requires that the DCG problem is solvable which made the problem meaningful.  Specifically, given all distributions, we can learn how the given components are combined. For example, if provided with images of various shapes and colors, we should be able to understand how shape and color interact to form specific images, such as a the image of red triangle. This assumption guarantees that there **exists** a way to recover these compositional rules from the data.
>
> > Q3: Are there plans to test the theory on synthetic?
>
> See W1
>
> > Q4: "composition rule" (T) for canonical DCG tasks?
>
> Composition rule is the rule of how the two components are combined. It can have different representation dependent on the data generative process. In image creation, the composition rule can be represented as  ``Draw the contour of a $<$shape$>$ and fill it with $<$color$>$.''. Similarly, for a robotic task, the composition rule can be expressed as “First, complete $<$subtask1$>$, then $<$subtask2$>$.”. The compositional rule can also be non-text representation. If the image is generated by a generative model, then the compositional rule can be represented as part of the generative model that focus on the combination of the two components. However, as long as these representation describe a same rule for component composition, we regard them as the different representation of a same composition rule.

---

> > ### Comment · Reviewer_PSGU · 2025-04-04
> >
> > Thanks for the authors, they solved my concerns. I raise the score to weak accept.

---

### Official Review · Reviewer_nkeW · 2025-03-14

**Overall Recommendation:** 4

**Summary:**

In this paper, the authors introduce a statistical framework to address two important research questions that have not been explored in prior work. Specifically, they examine whether a method designed for one DCG problem can be applied to another and identify the statistical properties that indicate a learning algorithm's capacity for knowledge composition

**Claims And Evidence:**

The theoretical claims in the paper are supported by proofs. However, it would be ideal to include some experimental validation, even in simple toy settings, to further substantiate the findings.

**Essential References Not Discussed:**

Essential references are discussed in detail, along with a thorough explanation of how the proposed framework differs from prior works.

**Experimental Designs Or Analyses:**

There are no experimental designs included. However, the analysis of the generalization bound, accounting for errors from two sources—insufficient data and knowledge composition—was examined. The analysis appears to be correct.

**Methods And Evaluation Criteria:**

Not applicable.

**Other Comments Or Suggestions:**

diagram (1) in Section 3.3 would benefit from a more detailed description.

**Other Strengths And Weaknesses:**

### Strengths
1. The paper addresses two important research questions in the area of DCG, and the proposed framework offers unique insights.
2. The paper is well-written, particularly the clear distinction between prior works and the current study. This effectively highlights the uniqueness of the proposed framework.
3. The analysis and theoretical framework presented in the paper will contribute to the development of better methods and provide a strong theoretical foundation for future research in this area.

### Weaknesses
1. There is a lack of experimental support or analysis, even in synthetic scenarios.
2. While the paper is generally well-written, it would be easier for readers to grasp the key ideas if the authors provided more intuitive explanations. For example, the paper introduces concepts like red rectangles and blue triangles, as well as Examples 3.2 and 3.3, but similar connections are missing in later sections. It would be beneficial if the authors explained major claims and conclusions from the theoretical study in a similar intuitive manner.
3. The practical implications of the proposed framework should be discussed in much more detail than in the current version.

**Questions For Authors:**

**Questions**
1 (Important). In the non-tradeoff scenario, the authors propose keeping the $\mu$ measure fixed and suggest data-centric approaches as a viable solution. Could the authors clarify how data-centric methods can effectively achieve this? Specifically, what should data-centric approaches focus on? Which aspects of training data development (such as collection, labeling, preparation, reduction, or augmentation) should practitioners prioritize for DCG, and what should be the key objectives during the data engineering phase?

2. Why were insufficient data and lack of knowledge composition considered the major factors contributing to generalization error?

**Relation To Broader Scientific Literature:**

In contrast to prior works that focus on DCG problems with specific composition rules, the proposed invariant measure explores the relationships between different composition rules. Additionally, the theorems presented are not confined to any particular learning problem. More importantly, the proposed theoretical framework emphasizes non-tradeoff improvements in DCG problems. This is the first paper to decouple the influence of finite samples and knowledge composition in generalization analysis for out-of-distribution settings. The proposed bound is tractable for various composition rules and establishes a connection between generalization behavior and mutual information.

**Theoretical Claims:**

The detailed proofs provided in the appendix were not reviewed, but all the claims presented in the main text were verified, and no issues were found.

---

> ### Author Rebuttal · Authors · 2025-03-31
>
> ## More background and examples
>
> ### Background:
> We will include a new section in the Appendix to provide additional background knowledge about **statistical machine learning**. The structure is:
>
> 1) Key concepts, including data space, learning algorithms, function space, and the i.i.d. (independent and identically distributed) assumption.
> 2) Generalization  analysis with Rademacher complexity as a example.
> 3) Current applications of generalization bounds .
> 4) extensionto out-of-distribution settings.
>
> ### Examples:
>
> **Example for Compositional rule**:  Composition rule is the rule of how the two components are combined. It can have different representation dependent on the data generative process. In image creation, the composition rule can be represented as  ``Draw the contour of a $<$shape$>$ and fill it with $<$color$>$.''. Similarly, for a robotic task, the composition rule can be expressed as “First, complete $<$subtask1$>$, then $<$subtask2$>$.”. The compositional rule can also be non-text representation, as long as it defines how two components are combined.
>
> **Example for inductive bias**: Inductive bias refers to a model's inherent preference for certain compositional rules before it is exposed to any training data for a given task. This bias can be introduced in two primary ways: Model Architecture Design – By carefully structuring the model, we can constrain its outputs to adhere to specific compositional rules. Pretraining & Objective Function – The inductive bias can also be shaped through pretraining strategies or the choice of objective function, either suppressing or reinforcing the model's tendency toward certain compositional behaviors.
>
> **Example for Assumption 5.3**: This assumption requires that the error is bounded. This assumption can be easily satisfied by modifying the original error using a $\min(\text{error}, \text{bound})$ operation. Alternatively, a bounded error measure, such as accuracy, which ranges between 0 and 1, can be used.
>
> **Example for Assumption 5.4**: This assumption ensures that, given all distributions, we can learn how the given components are combined. For example, if provided with images of various shapes and colors, we should be able to understand how shape and color interact to form specific images, such as an image of red triangle.
>
> **More examples** to demonstrate the indications of the theorem will be provided in the next version of paper.
>
> ## W & Q
>
> > W1: empirical validation
>
> See **''Experiments'' of rebuttal to Reviewer  XpR4**
>
> > W2: No examples
>
> See "More background and examples"
>
> > W3: Practical implication
>
> See Q1. More detail implication will be added in the next version of our paper.
>
> > Q1: Properties of data-centric methods and requirement for data engineering phase?
>
> Our theory suggests that a data-centric approach is fundamental for achieving non-trade-off improvements. It highlights the following key properties of data-centric methods:
>
> 1) Data-centric methods should effectively leverage information from the data itself. Injecting human task-specific knowledge into method design—such as using specialized model architectures or loss functions—may hinder the method's ability to learn directly from the data.
>
> 2) Theory 5.5 further asserts that compatibility between the learning algorithm and the data is crucial. This implies that the learning algorithm should achieve uniform performance across different compositions within the support distribution. For example, if the support distribution includes red triangles and blue rectangles, the model’s performance on red triangles and blue rectangles should be similar.
>
> Regarding the requirements for the data engineering phase, our theory supports the co-design of both the solution (including network structure and objective function) and data collection. Since the value of $I_{\mathcal{A},\beta}(\tilde{T}=T,P_S)$ in our theory depends on both the learning algorithm and the data, our theory cannot prescribe a universally optimal data development method independent of the specific approach. However, certain data quality requirements, such as the absence of label noise, are absolutely essential.
>
> > Q2: major factors contributing to generalization error
>
> In the DCG problem, the support distribution provides only a subset of knowledge combinations—for example, a red rectangle and a blue triangle. To generalize effectively, the model must first learn the concepts of colors and shapes and then recombine the learned concepts of "red" and "triangle" to form the concept of a "red triangle." Based on this property, we identify two major limiting factors: insufficient data and a lack of knowledge composition. Without sufficient data, the model tends to overfit to the small portion of training data, preventing it from learning the correct concepts. Meanwhile, the lack of knowledge composition indicates that the model struggles to recombine learned concepts, limiting its ability to generalize.

---

### Official Review · Reviewer_errU · 2025-03-24

**Overall Recommendation:** 3

**Summary:**

The authors analyze the problem of Distributional Compositional Generalization (DCG). Compositionality in this sense is the ability to model different features in the dataset and the statistical dependency between them. They try to provide statistical tools to assess whether it  is possible to transfer one  DCG problem to another and the capacity of the learning algorithm for DCG tasks.
They propose a measure (mu) which is based on the prediction function and invariant to the learning algorithm. They argue this measure can give insights on the DCG problem, with respect to different tasks and the respective learning algorithm.
The work is highly abstract and hard to follow. There are many notations and definitions, it appears the work is intended for a nich group who specializes in this topic and formulation. Although the authors try to give some intuition and examples words in the beginning, they do not go back to it later. There is no example, no illustration, not even a toy problem, let alone a real learning problem where the theory is applied.
Although some intresting insights may be learned here, I do not believe the venue of this conference is a good fit for this work.

**Claims And Evidence:**

see above

**Essential References Not Discussed:**

see above

**Experimental Designs Or Analyses:**

no experiments

**Methods And Evaluation Criteria:**

see above

**Other Comments Or Suggestions:**

More illustrations and intuition, connecting the more abstract notions to an actual learning problem.

**Other Strengths And Weaknesses:**

see above

**Questions For Authors:**

See above.

**Relation To Broader Scientific Literature:**

Relations to the more abstract literature. Do not review literature on current compositional representations and methods.

**Theoretical Claims:**

Several interesting theoretical claims.

---

> ### Author Rebuttal · Authors · 2025-03-30
>
> > W1:  There is no example, no illustration, not even a toy problem
>
> Actually, we provide the Example 3.2, Example 3.3 and an illustration in Appendix (Page 17).
>
> We have added more examples and background (detail **See ''More background and Examples'' of the rebuttal to Reviewer nkeW**) and experiments on a simulation problem (See **''Experiments'' of rebuttal to Reviewer  XpR4**).
>
>
>
> > W2: There are many notations and definitions, it appears the work is intended for a nich group who specializes in this topic and formulation.
>
> Our work presents the first statistical formalization of the DCG problems. We acknowledge that the paper introduces many notations and definitions, but this complexity reflects the inherent difficulty of the problem. To improve clarity, we have taken significant steps to simplify the presentation:
>
> 1)  We include schematic diagrams (Eq.1 and Eq.12) and provide the examples(Example 3.2 and Example 3.3).
> 2)  In the revised version, we have expanded the number of examples and added synthetic experiments to empirically validate our theoretical findings  (**see W1**).
>
> If you have any **concrete suggestions**, we would gladly incorporate them and revise the paper **promptly**. And this paper is **never** intended for a nich group — if you believe there are aspects that may inadvertently limit its accessibility, we would appreciate further details on which groups might find the presentation challenging.
>
> > W3: Do not review literature on current compositional representations and methods.
>
> Thank you for your suggestions.
>
> In the Related Work section, we already cover disentangled representation learning, which we view as a subset of compositional representation learning.
>
> And, to better address your point, we have expanded this section to include a dedicated discussion on current compositional representation learning methods, incorporating your suggested direction. Additionally, we will integrate the following literature into this new discussion:
>
> [1] Compositional Generalization in Unsupervised Compositional Representation Learning: A Study on Disentanglement and Emergent Language
>
> [2] CORL: Compositional Representation Learning for Few-Shot Classification
>
> [3] Representation Learning of Compositional Data
>
> [4] Measuring Compositionality in Representation Learning
>
> [5] Rule-Guided Compositional Representation Learning on Knowledge Graphs
>
> [6] THE ROLE OF DISENTANGLEMENT IN GENERALISATION
>
> [7] Where’s the Learning in Representation Learning for Compositional Semantics and the Case of Thematic Fit
>
> > W4: I do not believe the venue of this conference is a good fit for this work.
>
> I totally disagree with the point that the ICML is not suitable for our paper. The reason is given as follow:
>
> 1. This paper focuses on analyzing machine learning algorithms in DCG tasks, making it highly relevant to the topics listed in the ICML call for papers, specifically under "Theory of Machine Learning" (including statistical learning theory, bandits, game theory, decision theory, etc.).
> 2. ICML accepts many papers with dense theoretical analysis, and the other reviewers have provided relatively lengthy review comments, suggesting a certain level of interest in this paper in the ICML community.
> 3. ICML is an open and diverse community, which is one of the reasons I deeply appreciate it. We hope to uphold this openness. I acknowledge that theoretical researchers are somewhat in the minority within the community, but ICML has always provided opportunities for underrepresented perspectives.

---

> > ### Comment · Reviewer_errU · 2025-04-04
> >
> > In light of the rebuttal answers to my review and to the other reviewers, provided changes in the extended literature are given and additional examples and intuitions are added, I raise my rank for this paper.

---

### Official Review · Reviewer_XpR4 · 2025-03-25

**Overall Recommendation:** 4

**Summary:**

This paper proposes a theoretical statistical framework for analyzing Distributional Compositional Generalization (DCG). An invariant measure is proposed to evaluate the generalizability of methods across DCG tasks and derive a generalization bound separating the effects of insufficient data from knowledge composition capabilities. Their findings highlight the role of mutual information and algorithm-rule compatibility in DCG performance.

**Claims And Evidence:**

The claims presented in the paper are largely theoretical and mathematical. However, they lack empirical validation or experimental evidence to demonstrate their practical applicability and effectiveness. Specifically:
1. Although mathematically derived and proven, the claim regarding its practical utility or interpretability lacks empirical demonstration.
2. The theoretical bound separating data insufficiency from compositional errors is presented; however, the absence of empirical or simulation-based evidence makes it challenging to assess its practical tightness or usefulness.

Thus, while theoretically sound, the primary problematic claim is the applicability and practical relevance of these theoretical results without supporting empirical validation or concrete examples.

**Essential References Not Discussed:**

NA

**Experimental Designs Or Analyses:**

NA

**Methods And Evaluation Criteria:**

The proposed methods—namely, the invariant measure, i.e., invariant measure and the derived generalization bound, are conceptually appropriate and logically consistent with the theoretical objectives of understanding DCG. However, the paper lacks specific evaluation criteria or benchmark datasets to practically validate these theoretical contributions. Introducing empirical evaluation or clearly defined benchmarks would significantly strengthen the practical relevance and interpretability of the theoretical findings.

**Other Comments Or Suggestions:**

NA

**Other Strengths And Weaknesses:**

Check claims and evidence

**Questions For Authors:**

1. Could you provide examples or scenarios (even simulated) that empirically validate the theoretical invariant measure and generalization bound?
2. How do you envision your theoretical findings practically influencing algorithm design or improvement in real-world DCG tasks?
3. Could you explicitly discuss and justify the assumptions used in your generalization bound? Under what real-world conditions might these assumptions fail?

**Relation To Broader Scientific Literature:**

The paper extends statistical learning theory (e.g., PAC learning, NFL theorem) specifically to DCG. Unlike prior work focusing on specific DCG scenarios, it offers a general theoretical foundation.

**Theoretical Claims:**

Check claims and evidence

---

> ### Author Rebuttal · Authors · 2025-03-30
>
> ## 1. Experiments
>
> ### 1.1. Experiment Design:
>
> **1. Components and Compositional rule**： We construct two words set A,B satisfying |A|=|B|=1000 and their corresponding element $a_1,a_2\subset A$ and $b_1,b_2\subset B$. $a_1,a_2$ is a partition of $A$  and the same as $b_1,b_2$. $|a_1 |=|a_2|=|b_1|=|b_2|=500$.
> The composition rule can be any function that satisfy the following form:
> $(e_1,e_2)→e_1 e_2 e_1 e_2 e_1 e_1$.
> And we construct 64 composition functions, referred as  $T_1,T_2,\cdots,T_{64}$
>
> **2. Distribution Split** The support distribution takes the elements in the set $\lbrace(e_1,e_2)|(e_1,e_2 ) \in a_1 \times b_1 \cup a_2\times b_1 \cup a_1\times b_2 \rbrace$. The target distribution take elements in the set $\lbrace(e_1,e_2)|(e_1,e_2) \subset a_2\times b_2 \rbrace$. It is easy to verify that these designs satisfy the requirement listed in Section 3.
>
> **3. Sequence design** The input sequence is “$e_1,e_2,r_1,r_2,r_3,$#”, where $r_1, r_2, r_3$ are random words that simulate the randomness. The expected completed sequence is “$e_1,e_2,r_1,r_2,r_3,$#$,e_1,e_2,e_1,e_2,e_1,e_1$” if the composition rule is $(e_1,e_2)\rightarrow e_1,e_2,e_1,e_2,e_1,e_1$.
>
> **4. Learning algorithm design**
> In our paper, we define the learning algorithm as the mapping between data and the learned function, encompassing a broader concept than just the optimizer. To simulate learning algorithms with varying inductive biases and adaptivity, we adopt the following approach:
>
> 1. We employ the GPT-2 model with two configurations:
>    - **Setting 1:** 4 layers, 4 attention heads, and an embedding size of 128.
>    - **Setting 2:** 6 layers, 8 attention heads, and an embedding size of 256.
> 2. We pretrain the GPT-2 model using different pretraining data schedules. The pretraining data is generated from a subset of composition rules same to those in the downstream task, but with entirely different words. This setup allows us to create learning algorithms with different inductive biases and adaptivity while preventing data leakage.
>
>
> ### 1.2. Experiemnts on trade-off and non-trade-off improvement
>
> On of the key point in this paper is that the non-trade-off improvement has to rely on the adaptivity of learning algorithm (detail see beyond trade-off page 5). To verify this conclusion,  calculate $I_{\mathcal{A},\beta}(\tilde{T}=T,P_S)$, which is a measure of adaptivity used in out paper, and GACC, which is the average performance across all the tasks with compositional rule in $T_1,T_2,\cdots,T_{64}$. The results are:
>
>
> | $I_{\mathcal{A},\beta}(\tilde{T}=T,P_S)$ | 0.073 |	0.115 |	0.125 |	0.281 |	0.362 |	0.462 |	0.481 |	0.505 |	0.527 |	0.564 |
> | --- | -- | -- | -- | -- | -- | -- | -- | -- | -- | -- |
> | GACC | 0.605 |	0.591 |	0.565 |	0.633 |	0.696 |	0.750 |	0.772 |	0.789 |	0.752 |	0.776 |
>
>
>
>
>
> ### 1.3. Experiments on Generalization Bounds
>
> We conduct the experiments with different rule complexity using the best pretrain setting in previous section.
>
> Rule complexity refers to the length of the rule on the output side. For example, the rule complexity of $(e_1, e_2) \rightarrow e_1, e_2, e_1, e_2, e_1, e_1$ is 6, while the rule complexity of $(e_1, e_2) \rightarrow e_1, e_2, e_1, e_2, e_1, e_1, e_1, e_2$ is 8.
>
> | Rule Complexity | 6 | 8 | 10 | 12 |
> | --- | --- | --- | --- | --- |
> | CG Error | 0.223  | 0.262 | 0.301 | 0.342 |
> | Ben-David et al. | 0.622 | 0.680 | 0.701 | 0.690 |
> | Ours | 0.271 | 0.295 | 0.351 | 0.372 |
>
>
> The results indicate that our generalization bound is more tighter than the bound of Ben-David et al.
>
> ## 2. Question
>
> > Q1: empirical validation
>
> See 1.Experiments.
>
> > Q2: envision practical influence?
>
> See **Q1 in the response to Reviewer nkeW**
>
> > Q3: justify the assumptions
>
> Assumption 5.3 requires that the error is bounded. If the solution performs poorly on a small subset of data points but performs well on the rest, the average error could be disproportionately large due to extreme errors in that small subset without this assumption. This assumption can be easily satisfied by modifying the original error using a  $\min(\text{error}, \text{bound})$ operation. Alternatively, a bounded error measure, such as 1-accuracy, which ranges between 0 and 1, can be used.
>
> Assumption 5.4 requires that the DCG problem is solvable. Our bound does not apply to DCG problems that are entirely unsolvable. This assumption ensures that, given all distributions, we can learn how the given components are combined. For example, if provided with images of various shapes and colors, we should be able to understand how shape and color interact to form specific images, such as a the image of red triangle. This assumption guarantees that there **exists** a way to recover these compositional rules from the data.
>
> > w1: concrete examples.
>
> See **''More background and Examples'' of the rebuttal to Reviewer nkeW**

---

### Decision · Program_Chairs · 2025-05-01

**Decision:**

Accept (poster)

**Comment:**

All reviewers agree that the paper provides theoretical insights to better understand the generalization properties of DCG from statistical perspective. Initially, the reviewers had concern on the empirical support of the claims. The rebuttal was very helpful in clarifying questions raised. Based on the consistent positive reviews and the authors' rebuttal, AC recommends acceptance. Please make sure to include these clarifications as well as the extra results in the final version of the paper.